# Association of 25(OH)D serum level with biological aging: A Cross-Sectional Study of 2007–2016 NHANES surveys

Yi Feng[1], Licheng Yu[2], Jue Wang[1]*

1 Nanhui new town Community Health Service Center, Pudong New Area, Shanghai, China, 2 Prevention and Health Care Department, Heqing Community Health Service Center of Pudong New District, Shanghai, China

* wangjue180130@126.com

## Abstract

Vitamin D plays a crucial role in various aspects of human body, including aging, but chronological age may not accurately reflect the true biological aging status. Recently, PhenoAge has been developed to estimate an individual's biological age based on different biological and clinical measures. Therefore, we investigated the relationship between 25(OH)D serum levels and biological aging calculated by PhenoAge Acceleration (PhenoAgeAccel) using data from the 2007−2016 National Health and Nutrition Examination Survey (NHANES). 25(OH)D serum levels were negatively associated with PhenoAgeAccel (β = −0.04 standard deviation [SD]; 95% confidence interval [CI]: −0.08 to 0.00). This negative association was dose-dependent in females (β = −0.07 SD; 95% CI: −0.12 to 0.01), but not in males. Generalized additive models further revealed gender-specific non-linear patterns: a U-shape pattern in males but an L-shaped pattern in females. Using segmented regression to confirm inflection points, we observed that 25(OH)D serum levels were linked to reduced PhenoAgeAccel at levels below 38.2 nmol/L (15.3 ng/mL) in males (β = −0.013 SD; 95% CI: −0.025 to −0.002) and 62.5 nmol/L (25.0 ng/mL) in females (β = −0.007 SD; 95% CI: −0.01 to −0.004,). However, 25(OH)D serum levels above 125 nmol/L showed no association with PhenoAgeAccel in females (β = −0.001 SD; 95% CI: −0.005–0.002), while in males, elevated levels (>91.6 nmol/L) were associated with increased PhenoAgeAccel (β = 0.005 SD; 95% CI: 0.002–0.008). Our findings indicate that vitamin D insufficiency has an inverse link with accelerated biological aging, and high levels of vitamin D in males accelerated biological aging as well, offering valuable insights into the relationship between vitamin D and biological aging.

**Data availability statement:** Publicly available datasets were analyzed in this study. This data were from NHANES database (https://www.cdc.gov/nchs/nhanes). All relevant data from this database have been provided within the paper and its Supporting Information files.

**Funding:** This work was supported by the Special Clinical Research Project of Shanghai Municipal Health Commission in the form of a grant [20224Y0335 to YF].

**Competing interests:** The authors have declared that no competing interests exist.

## Introduction

Vitamin D is a fat-soluble vitamin that plays a crucial role in human health, influencing various functions such as bone metabolism and immune regulation. It is primarily obtained through two sources: endogenous production (mainly vitamin D3) in the skin upon exposure to ultraviolet B (UVB) radiations, and exogenous intake (vitamin D2 and D3) through dietary sources, such as fatty fish, egg yolks or supplements. Once absorbed in the body, vitamin D is converted into 25-hydroxyvitamin D (25(OH)D) in the liver, then transformed into the biologically active form, 1α,25-dihydroxyvitamin D (1α,25(OH)$_2$D), in the kidneys [1]. This active form binds to the vitamin D receptor (VDR) to regulate calcium and phosphate metabolism [2]. Serum 25(OH)D is the most commonly measured form of vitamin D in humans because it is the major circulating form and provides an accurate indication of the body's overall vitamin D status [3]; and can be influenced by various factors including the geographic location, race and food intake [4].

Vitamin D deficiency is a global public health concern, associated with increased risk of type 2 diabetes (T2D), hypertension, multiple sclerosis [5–7]. It is especially prevalent among the elderly [8], with some studies suggesting that deficiency may accelerate chronological aging and contribute to age-related diseases and inflammatory diseases, such as cardiovascular disease, multiple sclerosis and rheumatoid arthritis [9]. However, several large randomized clinical trials of vitamin D supplementation failed to demonstrate significant benefits for cardiovascular disease, casting doubt on the evidence from observational studies linking vitamin D deficiency with cardiovascular disease [10,11]. This ambiguity highlights challenges in assessing the association between vitamin D and aging. One major limitation of previous studies is that they relied on chronological age which does not always reflect an individual's biological age. Indeed, unlike chronological age which simply counts the years lived, biomarkers making up the biological age are mainly determined by a combination of genetic, environmental and lifestyle factors [12]. Consequently, some individuals may age biologically faster or slower than indicated by their chronological age.

To better characterize physiological aging in epidemiological studies, several aging metrics have been developed that combine clinical markers (e.g., PhenoAge and GrimAge) and biological (PhenoAge acceleration) markers [13]. In a post hoc analysis of the European DO-HEALTH randomized clinical trial involving 777 participants with a mean 25(OH)D serum level of 56.8 nmol/L, PhenoAge was the only biological age metric shown to be slowed by daily vitamin D (2000 UI) supplementation among the four biological age metrics that were assessed: PhenoAge, GrimAge, GrimAge2 and DunedinPACE [14]. PhenoAgeAccel was developed and validated using machine learning methods on a National Health and Nutrition Examination Survey (NHANES) dataset of 4000 adults aged 20–84 years, demonstrating a mean absolute error of approximately 3.6 years when predicting chronological age [15]. Although vitamin D supplementation has been shown to slow the increase of PhenoAgeAccel in older adults with low 25(OH)D serum concentrations, observational studies linking 25(OH)D serum concentrations with PhenoAgeAccel are limited. Therefore, this study aims

to investigate whether there is a linear and/or non-linear relationship between 25(OH)D serum concentration and Pheno-AgeAccel, using datasets of 2007–2016 NHANES surveys. By exploring this association, the study seeks to improve our understanding of the link between serum vitamin D levels and biological aging in human beings, with potential implications for public health.

## Materials and methods

### Study design and population

This study was conducted and reported in agreement with STROBE guidelines for reporting observational studies. This study utilized data derived from the 2007–2016 NHANES conducted in the United States of America by the National Center for Health Statistics of the Centers for Disease Control and Prevention. All data included in the current study are publicly available on the NHANES website [16]. NHANES includes in-home interview, physical measurements and laboratory tests. The 2007–2016 NHANES datasets involved 50558 adult participants. Participants with incomplete information and participants aged less than 40 years old were excluded. After applying the exclusion criteria, a total of 13,910 participants were finally included in this analysis (Fig 1).

### Measurement of 25(OH)D serum level

Vitamin D serum levels were reflected by 25(OH)D. Because the forms of 25(OH)D measured in the 2007–2016 NHANES samples included both 25(OH)D2 and 25(OH)D3 (classic 25(OH)D3 and C3 epimer of 25(OH)D3 [C3-epi-25(OH)D3]), serum 25(OH)D included 25(OH)D2, 25(OH)D3 and C3-epi-25(OH)D3 in this study. During the study period (2007–2016), 25(OH)D levels were measured in the NHANES population using the liquid chromatography-tandem mass spectrometry (LC-MS/MS). According to the 2011 Institute of Medicine's Committee recommendations, we transformed serum 25(OH)D concentrations into a categorical variable with the following modalities: deficiency (<30 nmol/L), insufficiency (30–50 nmol/L), sufficiency (50–75 nmol/L) and abnormally high (≥75 nmol/L) [17].

### Measurement of PhenoAge

In this study, we utilized PhenoAge as the indicator of biological aging, considering its high robustness and feasibility in NHANES data as a reliable predictor of mortality and age-related disease risks. The PhenoAgeAccel algorithm was

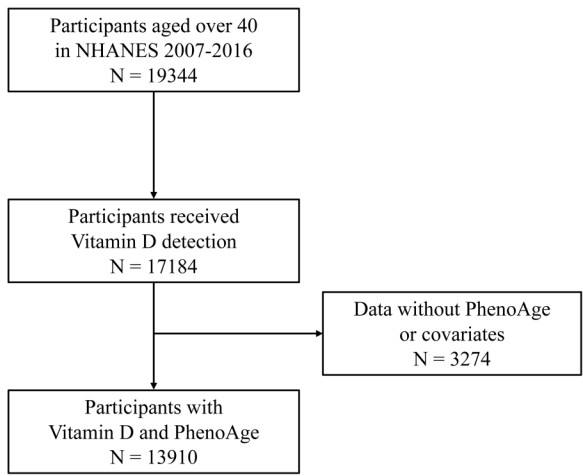

**Fig 1. Flowchart of participants selection.**

constructed by applying elastic-net regression methods to NHANES III (1988–1994) datasets, encompasses 10 markers: serum albumin, serum creatinine, plasma glucose, white blood cell count, blood lymphocyte percent, red cell distribution width, mean red cell volume, serum C-reactive protein, serum alkaline phosphatase and chronological age [18]. All constituent biomarkers of PhenoAge were routinely measured during NHANES using standardized protocols ensuring reproducibility across study cycles. One of the markers, CRP, was unavailable from 2011 to 2018 datasets. However, previous studies have applied an algorithm that excluded CRP when calculating the PhenoAge, demonstrating good applicability [19,20]. To assess the validity of PhenoAge without CRP data, we calculated 2 PhenoAge estimates with and without C-reactive protein, using data from cycles 2007–2008 and 2009–2010, which contained all 10 markers. We found a strong association between 2 PhenoAge estimates with and without CRP (Pearson's correlation coefficient of 0.986). Based on this finding, we proceeded with the main analysis of data from the five NHANES cycles using the PhenoAge algorithm excluding CRP. We further calculated PhenoAgeAccel as the difference between PhenoAge and chronological age. A positive value of PhenoAgeAccel represents accelerated biological aging process, while a negative value represented slower biological aging process. Subsequently, PhenoAgeAccel was standardized to have a mean of 0 and a standard deviation (SD) of 1.

## Covariates

NHANES interviews were conducted by trained interviewers. Questionnaires were administered to participants to collect information on demographics (chronological age, gender, race, education level, poverty income ratio (PIR) and lifestyle-related factors (tobacco smoking, alcohol consumption, physical activity, and comorbidities) and the body mass index (BMI). These variables were measured at the Mobile Examination Center.

Covariates in regression analyses testing the association between 25(OH)D serum levels and biological aging include age, race, education level, PIR, BMI, tobacco smoking, alcohol consumption, physical activity. The variable active tobacco smoking was classified into three categories: never smokers (previous smoking of a total of less than 100 cigarettes), current smokers (lifetime smoking of over 100 cigarettes) and former smokers (had smoked ≥100 cigarettes in their lifetime but had quit smoking at the time of the survey). The variable alcohol consumption was classified as never drinkers (participants who had drunk <12 drinks in their lifetime), current drinkers (participants who had drunk ≥12 drinks in their lifetime and at least one drink in the past year), and former drinkers (participants who had drunk ≥12 drinks in their lifetime but no drink in the past year). A Metabolic Equivalent task (MET)-min per week were calculated to assess physical activity, levels (in min per week) that were categorized as: inactive (no regular physical activity), insufficient (<500 MET-min activity/week), moderate (500–1,000 MET-min activity/week) and high (≥1,000 MET-min activity/week) [21]. For sleep duration, participants responded to the question: "How much sleep do you usually get at night on weekdays or on workdays?" Sleep duration was categorized into 3 groups: <7h, 7–8h and ≥8h [22]. The definition of hypertension was based on self-reports, the use of antihypertensive drugs or on the record of a systolic blood pressure≥140 mmHg with or without a diastolic blood pressure≥90 mmHg on physical exam [23]. The definition of diabetes mellitus was based on self-reports or on laboratory test results showing fasting blood glucose levels ≥126 mg/dL or HbA1c values ≥6.5% [24].

## Statistical analysis

Weighted multivariable linear regression models were constructed to estimate the association between serum 25(OH)D status and PhenoAgeAccel. 25(OH)D serum levels were log-transformed due to their skewed distribution. To evaluate whether the association between 25(OH)D serum levels and PhenoAgeAccel was not just linear but also dose-dependent, 25(OH)D serum level was analyzed as both a continuous (log-transformed) and a categorical variable. In agreement with standards for linear regression analyses involving continuous dependent variables, the association between 25(OH)D serum levels as continuous variables and PhenoAgeAccel was estimated based on beta coefficients with their 95% confidence intervals [25]. All models were adjusted for chronological age, gender, race, education level,

PIR, BMI, tobacco smoking and alcohol consumption variable, physical activity, sleep duration, hypertension, diabetes and climate season of measurement. To evaluate potential effect modification by gender and other key covariates, subgroup analyses and interaction tests were conducted. Interaction term of serum 25(OH)D and stratifying factor were included in the multivariable linear regression models to test for significance of interaction.

To explore whether there is also a non-linear relationship between 25(OH)D serum levels and PhenoAgeAccel, a Generalized Additive Model (GAM) was employed to fit the smooth curve. The GAMs extend traditional linear regression by replacing linear terms with smooth functions, allowing flexible modeling of complex dose-response relationships without assuming a priori parametric forms [26]. The threshold effect was further estimated using segmented regression model. This method identifies inflection points where the slope of the association between 25(OH)D serum levels and PhenoAgeAccel changes significantly. The optimal breakpoint was determined via likelihood ratio tests, comparing the goodness-of-fit between the non-segmented linear model and the segmented regression model [27]. Results with *P*-values <0.05 were considered to be statistically significance. All data analyses were performed using R software version 4.2.1 (R Core Team, Vienna, Austria). R package 'BioAge' was used to calculate PhenoAge [15], while the package 'mgcv' was used to conduct GAM analysis [28]. Continuous data were reported as means ± standard deviations (SDs) for normal distributions or as medians (interquartile ranges) for skewed distributions. Categorical data were reported as numbers (percentages).

### Ethical considerations

This study utilized data from NHANES study, which was conducted in accordance with the principles in the Declaration of Helsinki. The NHANES protocol received prior ethical approval from the National Center for Health Statistics' Review Board. Since our study is a secondary analysis of publicly accessible, de-identified NHANES data sets, additional institutional review board approval was not required [29].

## Results

### Characteristics of the study population

The current study included a total of 6836 males and 7074 females, with the characteristics of the study population presented in Table 1. Significant differences were observed between males and females in terms of age, race, BMI, and poverty-income ratio (p < 0.001). Additionally, gender-specific differences were noted in behaviors influencing biological aging, such as smoking, drinking, physical activity, and sleep duration (p < 0.001). Notably, males had a higher prevalence of diabetes mellitus than females. The median 25(OH)D serum level in females was 66.6 nmol/L compared with 63.8 nmol/L in males (p < 0.001). The PhenoAgeAccel of males and females were −2.40 (−5.13, 0.94) and −4.14 (−7.22, −0.46) respectively, suggesting that PhenoAgeAccel was lower in females than in males.

### Association between 25(OH)D serum level and biological aging

The associations between 25(OH)D serum levels and PhenoAgeAccel were presented in Table 2. After adjusting for age, race, education level, PIR, BMI, smoking, alcohol consumption, level of physical, sleep duration and presence of hypertension and/or diabetes mellitus, 25(OH)D serum levels were negatively associated with PhenoAgeAccel (β = −0.04 SD; 95% CI: −0.08 to 0.00). When stratified by sex, the association was statistically significant only in females (p = 0.007). Further analysis dividing serum 25(OH)D into quartiles revealed an inverse relationship with PhenoAgeAccel, although no clear dose-response trend was observed (p for trend = 0.173). When segregating the data by sex, 25(OH)D serum levels were inversely associated with PhenoAgeAccel only in women, with no apparent dose-response relationship (p for trend = 0.076). Categorizing 25(OH)D serum levels according to the 2011 USA institute of Medicine guidelines, inverse association with PhenoAgeAccel were observed for levels between 30–50 nmol/L and 50–75 nmol/L in males (without a clear dose-response trend), and between 50–75 nmol/L and ≥75 nmol/L in females, where a dose-response relationship

**Table 1. Characteristics of the study population.**

| Characteristics | Male (n = 6836) | Female (n = 7074) | P-value |
|---|---|---|---|
| Age (mean±SD) | 57.07 (11.3) | 58.12 (11.8) | <0.001 |
| Race | | | 0.001[b] |
| Non-Hispanic white | 3249 (47.5) | 3288 (46.5) | |
| Non-Hispanic black | 1360 (19.9) | 1417 (20.0) | |
| Others (Asian, Hispanic white, etc.) | 2227 (32.6) | 2369 (33.5) | |
| Education level | | | 0.394[b] |
| Below high school | 1847 (27.0) | 1831 (25.9) | |
| High school or equivalent | 1588 (23.2) | 1597 (22.6) | |
| Above high school | 3401 (49.8) | 3646 (51.5) | |
| Poverty-income ratio | | | <0.001[b] |
| ≤ 1.0 | 1199 (17.5) | 1447 (20.5) | |
| 1.0-3.0 | 2839 (41.5) | 3029 (42.8) | |
| > 3.0 | 2798 (40.9) | 2598 (36.7) | |
| BMI, kg/m$^2$ | | | <0.001[b] |
| < 25 | 1585 (23.2) | 1841 (26.0) | |
| 25-30 | 2749 (40.2) | 2117 (29.9) | |
| ≥ 30 | 2502 (36.6) | 3116 (44.1) | |
| Smoking | | | <0.001[b] |
| Never smoker | 2823 (41.3) | 4297 (60.7) | |
| Previous smoker | 2570 (37.6) | 1666 (23.6) | |
| Current smoker | 1443 (21.1) | 1111 (15.7) | |
| Drinking (alcohol consumption) | | | <0.001[b] |
| Never drinker | 527 (7.7) | 1572 (22.2) | |
| Previous drinker | 637 (9.3) | 1466 (20.7) | |
| Current drinker | 5672 (83.0) | 4036 (57.1) | |
| Level of physical activity | | | <0.001[b] |
| Inactive | 1775 (26.0) | 2533 (35.8) | |
| Insufficient | 802 (11.7) | 1086 (15.4) | |
| Moderate | 680 (10.0) | 842 (11.9) | |
| High | 3579 (52.4) | 2613 (36.9) | |
| Sleep duration | | | <0.001[b] |
| <7 h | 2572 (37.6) | 2445 (34.6) | |
| 7-8h | 1858 (27.2) | 1812 (25.6) | |
| ≥ 8 h | 2406 (35.2) | 2817 (39.8) | |
| Hypertension | | | 0.839[b] |
| No | 3007 (44.0) | 3067 (43.4) | |
| Yes | 3829 (56.0) | 4007 (56.6) | |
| Diabetes mellitus | | | <0.001[b] |
| No | 5086 (74.4) | 5515 (78.0) | |
| Yes | 1750 (25.6) | 1559 (22.0) | |
| 25(OH)D Serum level, nmol/L | | | |
| Median (IQR) | 63.8 (48.4, 80.2) | 66.6 (47.0, 87.3) | <0.001[c] |
| < 30 | 420 (6.1) | 545 (7.7) | <0.001[b] |
| 30-50 | 1451 (21.2) | 1457 (20.6) | |
| 50-75 | 2766 (40.5) | 2293 (32.4) | |
| ≥ 75 | 2199 (32.2) | 2779 (39.3) | |

*(Continued)*

  

**Table 1.** (Continued)

| Characteristics | Male (n = 6836) | Female (n = 7074) | P-value |
|---|---|---|---|
| PhenoAge (median [IQR]) | 53.8 (45.1, 64.6) | 53.1 (44.2, 64.1) | 0.032 |
| PhenoAgeAccel (median [IQR]) | −2.4 (−5.1, 0.9) | −4.1 (−7.2, −0.5) | <0.001[c] |
| Standardized PhenoAgeAccel (mean±SD) | 0.1 (0.9) | −0.2 (1.1) | <0.001[a] |

[a] Student's t-test was used to compare means of variables between two groups

[b] Chi-square test was used to compare categorical variables between two groups.

[c] Mann-Whitney test was used to compare non-normal variables between two groups.

was apparent (p = 0.014). Interaction test confirmed that association between serum 25(OH)D and PhenoAgeAccel was modified by gender, regardless of whether 25(OH)D was modeled as a continuous variable or categorized using quartiles or IOM guidelines (all p for interaction <0.001).

Results of subgroup/effect modification analysis showed that in males higher 25(OH)D serum levels were associated with reduction in PhenoAgeAccel primarily among individuals aged over 60 years, those with hypertension and those with diabetes mellitus, suggesting that there were statistically significant interactions between 25(OH)D serum levels and age, hypertension, and diabetes (Fig 2A). In females, no effect modifiers were identified in the association between 25(OH)D serum levels and PhenoAge Accel (Fig 2B).

## Association of 25(OH)D serum levels with PhenoAgeAccel on GAM

There was no linear association between serum 25(OH)D levels and PhenoAgeAccel when dividing the levels into quartiles (Table 2). Results from GAM confirmed a nonlinear association between 25(OH)D serum level and PhenoAgeAccel in both males and females (p < 0.001) (Fig 3A and 3B). In males, the relationship was U-shaped with the two inflection points at 38.2 nmol/L and 91.5 nmol/L. Specifically, low 25(OH)D serum levels (<38.2 nmol/L) were negatively related to PhenoAgeAccel (β = −0.013 SD; 95% CI: −0.025 to −0.002), whereas high levels (>91.55 nmol/L) showed a positive association (β = 0.005 SD; 95% CI: 0.002 to 0.008). In females, segmented regression models indicated an L-shaped relationship. 25(OH)D serum levels below 62.5 nmol/L were negatively linked to PhenoAgeAccel (β = −0.007 SD; 95% CI: −0.01 to −0.004), but no significant association observed beyond 125 nmol/L (β = −0.001 SD; 95% CI: −0.005 to −0.002). Overall, low (<38.2 nmol/L) or high (>91.6 nmol/L) 25(OH)D serum levels in males were associated with PhenoAgeAccel, while in females, only low levels (<62.5 nmol/L) were associated with PhenoAgeAccel (Table 3).

## Discussion

In the current study, the association between 25(OH)D serum level and biological aging was investigated in the 2007–2016 NHANES surveys. The analysis revealed a negative association between 25(OH)D serum levels and PhenoAgeAccel. Further effect modifier analysis showed that this association was influenced by age above 60 years, diabetes mellitus and hypertension, however, no such modifiers were identified for females. When subdividing 25(OH)D serum levels into quartiles and stratifying data by sex, this association was still observed only in females, and it exhibited a non-linear pattern. Additionally, when further subdividing 25(OH)D serum levels according to the Institute of Medicine in USA in 2011, the association persisted for 25(OH)D serum levels between 30–50 nmol/L and between 50–75 nmol/L in males and between 50–75 nmol/L and ≥75 nmol/L, with a linear pattern of relationship observed only in females. Non-linear associations were also found between 25(OH)D serum levels and PhenoAgeAccel in both males (U-shaped) and females (L-shaped).

Previous studies have reported that a gradual decline in 25(OH)D serum levels with chronological age, and vitamin D supplementation can mitigate oxidative damage and improve immune function in elder people, suggesting its potential role

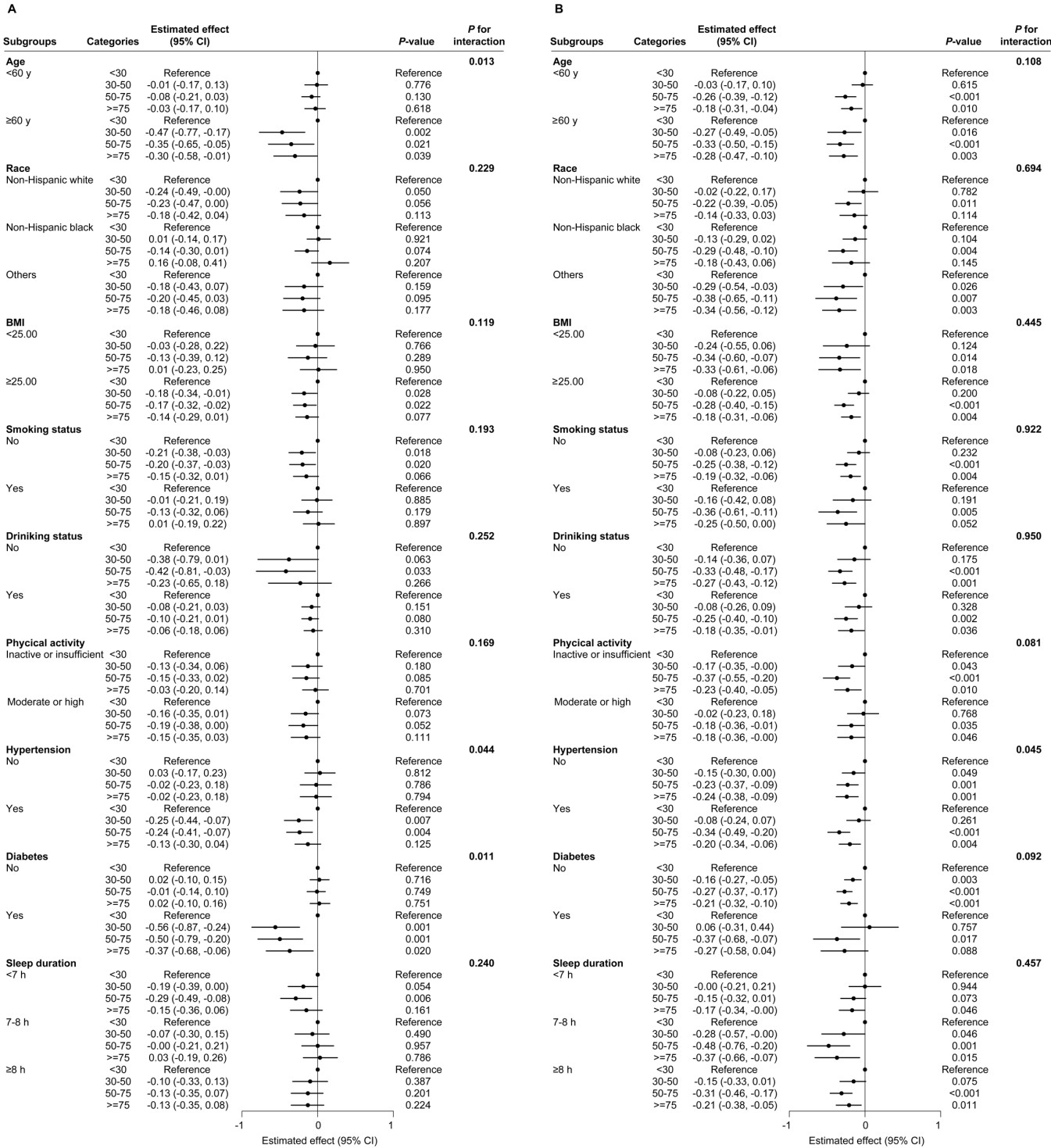

**Fig 2. Subgroup analyses of the association between 25(OH)D categories (based on the 2011 Institute of Medicine) and PhenoAgeAccel among males.** (A) and females (B). The filled circles and horizontal lines represent regression coefficients and the 95% confidence intervals. Multiple linear regression models were adjusted for age, race, education level, poverty-income ratio, body mass index, smoking, drinking, physical activity, sleep duration, hypertension, diabetes and season. BMI, body mass index; CI, confidence intervals.

Table 2. Association between 25(OH)D serum level and PhenoAgeAccel.

| Characteristics | All | | | Male | | | Female | | |
|---|---|---|---|---|---|---|---|---|---|
| | Beta coefficient (95% CI) | | P-value | Beta coefficient (95% CI) | | P-value | Beta coefficient (95% CI) | | P-value |
| | Crude model | Adjusted model [a] | | Crude model | Adjusted model [a] | | Crude model | Adjusted model [a] | |
| Log(25(OH)D) | −0.27 (−0.32, −0.22) | −0.04 (−0.08, 0.00) | 0.039 | −0.13 (−0.20, −0.06) | 0.03 (−0.02, 0.09) | 0.334 | −0.32 (−0.38, −0.26) | −0.07 (−0.12, −0.01) | 0.007 |
| Quartiles | | | | | | | | | |
| Q1 | Reference | Reference | | Reference | Reference | | Reference | Reference | |
| Q2 | −0.23 (−0.29, −0.17) | −0.10 (−0.16, −0.05) | <0.001 | −0.16 (−0.24, −0.08) | −0.02 (−0.09, 0.04) | 0.448 | −0.33 (−0.41, −0.25) | −0.16 (−0.23, −0.08) | <0.001 |
| Q3 | −0.29 (−0.36, −0.23) | −0.10 (−0.16, −0.04) | 0.001 | −0.20 (−0.29, −0.11) | −0.03 (−0.12, 0.04) | 0.355 | −0.40 (−0.49, −0.32) | −0.15 (−0.22, −0.07) | <0.001 |
| Q4 | −0.32 (−0.38, −0.26) | −0.05 (−0.11, −0.00) | 0.022 | −0.16 (−0.24, −0.07) | 0.02 (−0.04, 0.10) | 0.553 | −0.42 (−0.51, −0.34) | −0.10 (−0.18, −0.02) | 0.009 |
| P for trend | | | 0.173 | | | 0.429 | | | 0.076 |
| Categories based on the 2011 Institute of Medicine | | | | | | | | | |
| <30 | Reference | Reference | | Reference | Reference | | Reference | Reference | |
| 30-50 | −0.23 (−0.33, −0.13) | −0.14 (−0.23, −0.05) | 0.002 | −0.26 (−0.41, −0.11) | −0.15 (−0.29, −0.02) | 0.024 | −0.22 (−0.34, −0.09) | −0.10 (−0.21, 0.00) | 0.057 |
| 50-75 | −0.42 (−0.52, −0.33) | −0.21 (−0.30, −0.13) | <0.001 | −0.39 (−0.53, −0.25) | −0.16 (−0.29, −0.04) | 0.009 | −0.51 (−0.63, −0.40) | −0.24 (−0.33, −0.15) | <0.001 |
| ≥75 | −0.46 (−0.56, −0.36) | −0.15 (−0.24, −0.06) | 0.001 | −0.36 (−0.50, −0.22) | −0.09 (−0.22, 0.03) | 0.131 | −0.54 (−0.66, −0.42) | −0.17 (−0.27, −0.07) | 0.001 |
| P for trend | | | 0.130 | | | 0.366 | | | 0.014 |

[a]Adjusted for age, race, education level, poverty-income ratio, body mass index, smoking, drinking, physical activity, sleep duration, hypertension, diabetes and season of measurement.

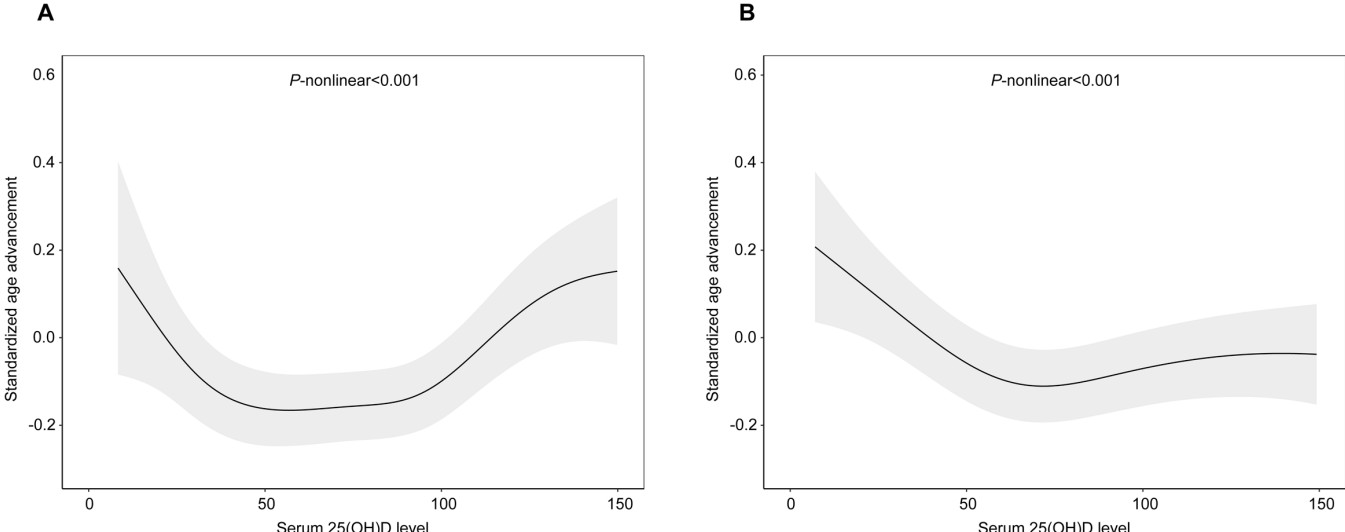

**Fig 3. Nonlinear relationships between 25(OH)D serum level and PhenoAgeAccel among males (A) and females (B).** Generalized additive models were adjusted for age, race, education level, poverty-income ratio, body mass index, smoking, drinking, physical activity, sleep duration, hypertension, diabetes and season. The black solid line represents the smooth curve fit between variables. The grey band represents the 95% confidence interval of the fit.

in slowing age-related physiological decline [30]. This decline in vitamin D levels is attributed to multiple factors, including decreased sun exposure, reduced capacity of the skin to synthesize vitamin D, and decreased dietary intake of vitamin D, all of which contribute to vitamin D deficiency in older adults [31]. The synthesis of vitamin D from sun exposure vary in different races, but our study lacks this data of sun exposure. Additionally, we also found that low 25(OH)D serum levels were associated with positive PhenoAgeAccel, suggesting vitamin D may play a crucial role in the regulation of the aging process. Adequate 25(OH)D levels are essential for maintaining calcium absorption and bone health preventing osteoporosis, a common age-related condition, and regulating calcium homeostasis and bone metabolism [32]. Beyond these functions, vitamin D can interact with antioxidant and anti-inflammatory pathways, such as activating Nrf2 (nuclear factor erythroid 2-related factor 2) and SIRT1 (sirtuin 1) signaling cascades, and suppressing the production of pro-inflammatory cytokines, including interleukin-6 (IL-6) and tumor necrosis factor-alpha (TNF-α), through inhibiting nuclear factor-κB (NF-κB). These

**Table 3. Results of the analysis of the relationship between 25(OH)D serum levels and PhenoAgeAccel.**

| 25(OH)D serum levels (nmol/L) | Beta coefficient (95% CI) | *P*-value |
|---|---|---|
| Male | | |
| <38.2 | −0.013 (−0.025, −0.002) | 0.017 |
| 38.2-91.6 | 0.001 (−0.001, 0.002) | 0.487 |
| ≥91.6 | 0.005 (0.002, 0.008) | <0.001 |
| Female | | |
| <62.5 | −0.007 (−0.010, −0.004) | <0.001 |
| 62.5-125.0 | 0.002 (0.000, 0.004) | 0.029 |
| ≥125.0 | −0.001 (−0.005, 0.002) | 0.535 |

Generalized additive models were adjusted for age, race, education level, poverty-income ratio, body mass index, smoking, drinking, physical activity, sleep duration, hypertension, diabetes and season.

actions help reduce inflammation and oxidative stress, both of which are known contributors to the aging process [32,33]. Consequently, vitamin D and aging influence each other mutually.

Vitamin D supplementation has been shown to be an effective way to reverse or mitigate diseases, including lower blood pressure and improve endothelial function, leading to reduced risk of cardiovascular events, such as heart attacks and strokes, and improved cardiovascular outcomes [34]. A meta-analysis of 75 randomized controlled trials in 2014 reported that vitamin D supplementation was associated with a 6% reduction in all-cause mortality [35]. Notably, the impact of vitamin D supplementation on mortality appears more significant in certain subgroups, such as older adults or individuals with vitamin D deficiency. For instance, vitamin D supplementation was associated with a reduced risk of all-cause mortality in trials where a mean baseline 25(OH)D level were below 75 nmol/L (RR 0.89, 95% CI 0.80–0.99), but not in trials with a mean baseline 25(OH)D level ≥75 nmol/L (RR 1.04, 95% CI 0.97–1.11) [36]. Thus, these evidence highlights the potential of vitamin D supplementation in reducing the risk of several chronic diseases, particularly in the elderly, in whom serum vitamin D levels tend to decline with age.

Despite its potential benefits, the use of vitamin D supplements has increased markedly, raising concerns about the risk of excessive intake. The vitamin D and Omega-3 Trial (VITAL) study, a randomized trial involving over 25,000 participants, discovered that vitamin D supplementation did not significantly reduce the risk of all-cause mortality or lower risk of fractures among generally healthy midlife and older adults without vitamin D deficiency or osteoporosis [37]. Furthermore, increased intake of vitamin D supplements by the general population and a growing number of high-dose prescriptions without medical monitoring might result in a greater risk of excessive vitamin D toxicity [38]. Our current results indicated that the association of 25(OH)D serum level and biological aging followed a U-shaped curve in males and an L-shaped curve in females. This suggests that high serum level of 25(OH)D is not associated with slower aging and may, in fact, be linked to accelerated biological aging in males. These detrimental effects may stem from the toxicity of excessive vitamin D. Excessive intake can cause serum levels of its metabolites, such as 25(OH)D and $24,25(OH)_2D$, to surpass the binding capacity of vitamin D binding protein, leading to an increase in free, active $1,25(OH)_2D$. This active form reaches the VDR in the nucleus, constantly promoting gene expression that enhances calcium absorption and bone mobilization [39]. Consequently, this can result in hypercalcemia or hyperphosphatemia, which in turn suppresses initial parathyroid hormone (iPTH) level [39,40]. Such disruptions in calcium balance and hormone regulation may contribute to cardiovascular disease, kidney damage, and skin aging. Research reported that the working diagnosis of vitamin D toxicity is diagnosed based on total 25(OH)D concentrations of 80～200nmol/L detected by immunoassay [41]. Our findings that the 25(OH)D levels over 91.5 nmol/L in male participants were not linked to accelerated biological aging are approximately consistent with those reports.

Our study identified a transition point in the relationship between 25(OH)D serum level and biological aging, with males exhibiting a threshold at 38.20 nmol/L and females at 62.50 nmol/L. Beyond this transition point, increase in 25(OH)D level no longer associates with biological aging deceleration. Regarding the threshold of vitamin D deficiency, it remains an ongoing controversy. The Scientific Advisory Committee on Nutrition (2016) recommended threshold at 25 nmol/L among UK population [42], whereas the IOM (2011) suggested a threshold of 30 nmol/L based on evidence in U.S. and Canadian populations [17]. Meanwhile, the European Food Safety Authority 2016 and the Endocrine Society guideline 2011 proposed higher level of 50 nmol/L as deficiency cutoff [43,44]. The discrepancy in cutoff values stems from multiple factors, including different regional ultraviolet B exposure patterns, dietary practice and population-specific biological variability. Furthermore, guideline variations also reflect differing priorities concerning health outcomes, with some prioritizing musculoskeletal health [45]. Notably, in 2024, the Endocrine Society guideline retracted their previous recommendation of threshold at 50 nmol/L due to limited clinical trial evidence [46]. They also highlighted the impracticality of universal thresholds due to heterogeneity in health outcomes and population characteristics. Therefore, the interpretation of our findings regarding 25(OH)D level threshold should be with caution, especially since it derived from observational study in an ethnically diverse population. Nevertheless, the trend of benefits attenuation

we observed is worth noting, suggesting excessive level of serum 25(OH)D may not be beneficial regarding biological aging process.

Our results revealed sex-specific differences of dose-response relationship between serum 25(OH)D levels and biological aging, which may be attributed to the sex hormones, particularly estrogen. As women age, especially during the menopausal transition and after menopause, their ovaries produce less estrogen, leading to a decline of estrogen levels. This decline contributes to the thinning and wrinkling of the skin, as well as changes in hair texture and growth patterns [47]. Some studies suggest that vitamin D could enhance the activity of estrogen receptors (ER) via interacting with them, potentially amplifying the anti-aging effects of estrogen in the body [48]. Additionally, vitamin D and estrogen also have a synergistic effect on bone metabolism, helping to prevent osteoporosis and fractures, especially in postmenopausal women [49]. Vitamin D may also regulate estrogen metabolism, leading to higher circulating estrogen levels and subsequently delay aging [48]. Therefore, high level of serum 25(OH)D may not be associated with decelerated biological aging in females. However, the mechanisms underlying the adverse effects observed at extremely high levels (>100 nmol/L) in males remain unclear. Further studies are needed to clarify the complex interplay between these two important hormonal systems.

This study has several limitations. First, as a cross-sectional investigation, it only assesses association between serum 25(OH)D levels and biological aging, making it challenging to establish the causality. Second, the classification of 25(OH)D serum levels used in our study is based on the 2011 IOM guidelines; however, more recent classifications, such as those from the Scientific Advisory Committee on Nutrition (2016) [50], may provide more accurate benchmarks. Analyzing the relationship using these updated criteria would be preferable. While sun exposure and dietary intake are primary sources of vitamin D, data on these factors, as well as supplement use, were not included in our analysis due to the lack of corresponding data in the NHANES dataset. The absence of these confounding variables may introduce residual confounding, reverse causation, and bias. Additionally, race significantly influences serum vitamin D levels, but the definition of race varies across NHANES data collected over different years, which posed a challenge in our study. Finally, emerging evidence questions the traditional view of serum 25(OH)D as a definitive biomarker of vitamin D status, although it remains the most widely accepted marker. For instance, in individuals with diabetes or obesity, the expression of CYP2R1—the enzyme responsible for vitamin D 25-hydroxylation—is suppressed, leading to decreased 25(OH)D levels despite stable overall vitamin D levels [51,52].

## Conclusion

In this study, we investigated the relationship between 25(OH)D serum levels and PhenoAge in NHANES, and found that there was a negative association between PhenoAgeAccel and 25(OH)D serum level, and these associations were nonlinear in both genders, revealing a U-shaped curve for males and an L-shaped curve for females. Low 25(OH)D serum levels shows a potential positively link to PhenoAgeAccel in both genders, but high 25(OH)D serum levels do not associate with decelerated PhenoAge, and even associated with accelerated PhenoAge in male. Our results suggest that the association between vitamin D levels with biological aging is different in men and women.

## Supporting information

**S1 Table. STROBE checklist.**
(DOCX)

**S1 File. Statistical analysis plan.**
(DOCX)

**S2 File. Dataset.**
(CSV)

## Author contributions

**Conceptualization:** Yi Feng.

**Data curation:** Yi Feng, Licheng Yu.

**Formal analysis:** Yi Feng, Licheng Yu.

**Methodology:** Yi Feng.

**Project administration:** Jue Wang.

**Supervision:** Jue Wang.

**Writing – original draft:** Yi Feng.

**Writing – review & editing:** Jue Wang.

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
