## [Decision Letter · Decision Letter 0]

9 Feb 2025

Dear Dr. Wang,

Thank you for submitting your manuscript to PLOS ONE. After careful consideration, we feel that it has merit but does not fully meet PLOS ONE publication criteria as it currently stands. Therefore, we invite you to submit a revised version of the manuscript that addresses the points raised during the review process.

We look forward to receiving your revised manuscript.

Kind regards,

Mickael Essouma, M. D.

Academic Editor

PLOS ONE

Journal Requirements:

Additional Editor Comments:

In addition to the reviewer's comments, I have the following comments:

1. In the title, you would better replace "age advancement" with "biological aging" or "physiological aging" given that PhenoAge used as metric of aging in this study is a measure of biological/physiological aging.

2. The authors should revise the "Materials and methods", results and discussion sections based on the STROBE guidelines statement (see von Elm et al. PLoS Med. 2007;4(10):e296) to allow for the reproducibility of their methods. Accordingly, they should start the "Materials and methods section" stating that their report conforms to STROBE guidelines for observational studies, and upload a STROBE checklist in the appendix. The "Statistical analysis" sub-section should follow the SAMPL guideline statement (see Lang TA et al. Medical Writing.2016;25(3):31-36) and the authors should upload a statistical analysis plan for this study as well as raw data (if legal frameworks of NHANES allow that) as supplemental materials to this manuscript. This recommendation aims to improve the readability of their manuscript, allow the credibility and reproducibility of their study methods, and increase the credibility of their findings and trustworthiness of the interpretation of their findings.

3. The "Materials and methods" section should end with a sub-section titled "Ethical considerations"in which the authors should state whether they abode by the ethical rules for conducting medical/health research (see the 2024 declaration of Helsinki statement published in JAMA), how they accessed NHANES (USA) data whereas all authors are affiliated with China institutes and the study was funded by a Chinese organization). Did they get an IRB approval to use the NHANES dataset?

4. In the conflicts of interest statement on the manuscript's agenda, the authors should clarify any potential conflict of interest with the funder and/or NHANES database managers.

5. It is difficult to know in tables and figures what type of effect magnitude estimates (odds ratios? Others?) the authors base their conclusions on and whether those estimates are those that should be used in cross-sectional studies to assess associations: see Thiese MS. Observatiobal and interventional study design types; an overview. Biochemistry Med (Zagreb).2014;24(2):199-210. Did you perform logistic regression analyses and/ or correlation analyses? Why perform both or why perform one of them? These clarifications need to be made in the statistical analysis sub-section of the "Materials and methods" section. Along these lines, I would like to remember that the authors overstated their conclusions (see Schooling MC and Heidi JE. Clarifying questions about "risk factors": predictors versus explanation. Epidemiology in Themes. 2018;15:10), and this should be corrected throughout the manuscript.

6. The authors cite res 4 to 6 for the definition of vitamin D deficiency but use ref 18 for the definition of vitamin D deficiency in this study. Why can't they use only reference 6 in the introduction and for the definition in this study given that it is the most recently published article? Can you specify whether you assessed the human form of 25-OH-D (25-OH-D3) throughout the manuscript?

7. After reading your introduction, I still do not well understand the rationale for conducting this study given that as well mentioned in reference 6 of the current manuscript, it is largely known that older individuals generally have lower vitamin D levels than younger populations. Do you want to asses whether that knowledge based on social age could hold even when using biological aging metrics (taking into consideration the discrepancy between social and biological ages)? The aim and rationale for conducting the study should be unambiguous in the last paragraph of the introduction. Still regarding the introduction, it is important to begin to be more cautious and specific when making claims about the involvement of vitamin D in human disease based on epidemiological data (see Vitamin D and human healrh: evidence from Mendelian randomization studies. Eur J Epidemiol. 2024;39: 467-490 and Schooling MC and Jones HE. Clarifying questions about "risk factors": predictors versus explanation. Epidemiology in Themes. 2018; 15:10). From my perspective, in a study whereby you seemingly assess the epidemiological link between vitamin D and physiological aging, the introduction should (in a brief overview form: introduction of no more than 1.5 pages) highlight how the human form of vitamin D evolves throughout the age spectrum based on the most certain available peer-reviewed evidence, as well as what is known and what is unknown about the epidemiological association between the human form of vitamin D and physiological aging, before stating the rationale for conducting the study (gaps in knowledge you would like to fill) and the study's aim.

8. Finally, language editing is also required.

Reviewers' comments:

Reviewer's Responses to Questions

**Comments to the Author**

1. Is the manuscript technically sound, and do the data support the conclusions?

Reviewer #1: Partly

2. Has the statistical analysis been performed appropriately and rigorously?

Reviewer #1: N/A

3. Have the authors made all data underlying the findings in their manuscript fully available?

Reviewer #1: Yes

4. Is the manuscript presented in an intelligible fashion and written in standard English?

Reviewer #1: No

Reviewer #1: Feng et al conduct this research in a rare direction while most of researchers will focus on the association between 25(OH)D and disease. Here are some issues should be concerned.

1. In the abstract, there is no statistical results. The conclusion that VD deficiency promotes aging is not proper since cross sectional study only remind the possible association between exposure and outcome instead of causuality.

2. The introduction section is a little long, in which 3-4 paragraphes can be better.

3. In methods section, data including flow chart should be added, and please learn some from the below articles:

a. Possible non-linear relation between prostate specific antigen and vitamin D: a machine learning study based on cross-section data

b. Association between urinary metals and prostate-specific antigen in aging population with depression: a cross-sectional study

4. In the discussion section, this study should declare the mechanism in the increase of 25(OH)D acclerate aging.

5. Except the increase of 25(OH)D acclerate aging, if there are some literatures reporting aging can increase 25(OH)D. Maybe it is hard to explain the causality of the interactive association between 25(OH)D and aging. Please make this part clear.

6. Language should be revised. Like in conclusion, "These relationships exhibited nonlinearity in both males and females.". It is not clear between what.

**Do you want your identity to be public for this peer review?** For information about this choice, including consent withdrawal, please see our Privacy Policy

Reviewer #1: No

---

## [Author Response · Author response to Decision Letter 1]

20 Mar 2025

Dear editor and reviewer:

Thank you very much for your careful review and constructive suggestions with regard to our manuscript. Those comments are helpful for us to revise and improve our paper. We have studied comments carefully and tried our best to revise and improve the manuscript and made great changes in the manuscript according to the good comments. We earnestly appreciate for the warm work from you, and hope that the corrections will meet with approval. Please feel free to contact us with any questions and we are looking forward to your consideration. The main corrections in the paper and the responds to your comments are as following:

Additional Editor Comments:

In addition to the reviewer's comments, I have the following comments:

1. In the title, you would better replace "age advancement" with "biological aging" or "physiological aging" given that PhenoAge used as metric of aging in this study is a measure of biological/physiological aging.

Response: Thank you for your insightful feedback. We have revised the title to use "biological aging" instead of "age advancement" to more accurately reflect our study's focus on the biological aspects measured by the PhenoAge metric.

2. The authors should revise the "Materials and methods", results and discussion sections based on the STROBE guidelines statement (see von Elm et al. PLoS Med. 2007;4(10):e296) to allow for the reproducibility of their methods. Accordingly, they should start the "Materials and methods section" stating that their report conforms to STROBE guidelines for observational studies, and upload a STROBE checklist in the appendix. The "Statistical analysis" sub-section should follow the SAMPL guideline statement (see Lang TA et al. Medical Writing.2016;25(3):31-36) and the authors should upload a statistical analysis plan for this study as well as raw data (if legal frameworks of NHANES allow that) as supplemental materials to this manuscript. This recommendation aims to improve the readability of their manuscript, allow the credibility and reproducibility of their study methods, and increase the credibility of their findings and trustworthiness of the interpretation of their findings.

Response: Thank you for your constructive suggestions and guidance. We have thoroughly revised the manuscript to ensure alignment with STROBE guidelines for observational studies, as well as the SAMPL recommendations for statistical reporting. Meanwhile, in the Materials and methods section, we supplemented the following statement: “This study was conducted and reported in adherence to STROBE guidelines for observational studies.” (marked in blue, Page 4, Lines 95). The completed STROBE checklist is provided in Supplementary materials. A statistical analysis plan has also been written and provided in Supplementary materials. Regarding data accessibility, all raw data was publicly available through the NHANES website: https://wwwn.cdc.gov/nchs/nhanes/Default.aspx. According to the NHANES data use agreement, redistribution of raw datasets is prohibited. Therefore, we are unable to provide NHANES raw data directly. This has been clarified in the Materials and methods section: “All data included in current study are publicly available through the NHANES website [18].” (marked in blue, Page 5, Lines 99).

The following reference was supplemented in the References List:

19. Centers for Disease Control and Prevention National Health and Nutrition Examination Survey Questionnaires D, and Related Documentation 2020. [cited 2025 March 7]. Available from: https://wwwn.cdc.gov/nchs/nhanes/.

3. The "Materials and methods" section should end with a sub-section titled "Ethical considerations"in which the authors should state whether they abode by the ethical rules for conducting medical/health research (see the 2024 declaration of Helsinki statement published in JAMA), how they accessed NHANES (USA) data whereas all authors are affiliated with China institutes and the study was funded by a Chinese organization). Did they get an IRB approval to use the NHANES dataset?

Response: We thank the editor for your constructive suggestions and guidance. Our research adhered to the ethical principles in the 2024 Declaration of Helsinki regarding secondary analysis. It exempts studies from additional ethical review when utilizing pre-existing de-identified data [1]. Meanwhile the NHANES datasets we used were publicly available and did not contain restricted use datasets. According to the CDC/NCHS Research Ethics Review Board statement, researchers using only the publicly available data files do not need to seek IRB approval [2]. Hence, no additional IRB was required in our study.

In the Materials and methods section, we have supplemented the “Ethical considerations” sub-section in the revised manuscript: “This study utilized data from NHANES study, which was conducted in accordance with the principles in the Declaration of Helsinki. The NHANES protocol received prior ethical approval from the National Center for Health Statistics' Review Board. Since our study is a secondary analysis of publicly accessible, de-identified NHANES data sets, additional institutional review board approval was not required [26, 27].” (marked in blue, Page 7, Lines186).

The following reference was supplemented in the References List:

26. University of Connecticut. Guidance on secondary analysis of existing data sets. [cited 2025 Feb 5]. Available from: https://ovpr.uconn.edu/services/rics/irb/researcher-guide/secondary-analysis-of-data-sets/.

27. World Medical A. World Medical Association Declaration of Helsinki: Ethical Principles for Medical Research Involving Human Participants. JAMA. 2025;333(1):71-4. Epub 2024/10/19 20:42. doi: 10.1001/jama.2024.21972. PubMed PMID: 39425955.

4. In the conflicts of interest statement on the manuscript's agenda, the authors should clarify any potential conflict of interest with the funder and/or NHANES database managers.

Response: We thank the editor for your guidance. We have revised the “Conflict of interest” statement in the manuscript as follows: “The authors declare no competing financial or non-financial interests related to this work. NHANES is a publicly available dataset managed by the Centers for Disease Control and Prevention. The authors confirm that there is no affiliation with or involvement from the NHANES program administrators in this study. The funding source had no role in study design, data collection, analysis, or manuscript preparation.”.

5. It is difficult to know in tables and figures what type of effect magnitude estimates (odds ratios? Others?) the authors base their conclusions on and whether those estimates are those that should be used in cross-sectional studies to assess associations: see Thiese MS. Observatiobal and interventional study design types; an overview. Biochemistry Med (Zagreb).2014;24(2):199-210. Did you perform logistic regression analyses and/ or correlation analyses? Why perform both or why perform one of them? These clarifications need to be made in the statistical analysis sub-section of the "Materials and methods" section. Along these lines, I would like to remember that the authors overstated their conclusions (see Schooling MC and Heidi JE. Clarifying questions about "risk factors": predictors versus explanation. Epidemiology in Themes. 2018;15:10), and this should be corrected throughout the manuscript.

Response: We sincerely thank the editor for your guidance. Regarding the choice of effect estimates, our analyses utilized beta coefficients derived from weighted multivariable linear regression models. According to Thiese et al., odds ratios are standard for binary outcomes in logistic regression. However, when measuring continuous response variables, beta coefficients are recommended as they quantify linear association between continuous variables. To improve the clarity of our manuscript, we supplemented the following sentences in the Materials and methods section: “Beta coefficients with 95% confidence intervals (CIs) were reported, representing the change in dependent variable per unit increase in the independent variable. The use of beta coefficients provides interpretable estimates for linear associations when analyzing continuous dependent variables, which is widely recommended in epidemiological studies.” (marked in blue, Page 7). Meanwhile, we made adjustment to all Tables and Figures, as well as description of results to ensure the use of beta coefficient is straightforward.

We also supplemented the following sentences in the Materials and methods section to explain why correlation analysis is omitted in current study: “Correlation analysis was not employed because it does not account for confounding factors, whereas the primary aim of current study is to estimate the confounder-adjusted association between 25(OH)D status and PhenoAge aging.” (marked in blue, Page 7).

The following reference was supplemented in the References List:

24. Bender R. Introduction to the use of regression models in epidemiology. Cancer epidemiology: Springer; 2009. P. 179-95

Regarding the potential overstatement of conclusion, we have revised the manuscript following the guidance of Schooling and Heidi (2018). Adjustments have been made throughout the Abstract, Results, and Discussion and Conclusion sections to avoid formative and causal language.

6. The authors cite res 4 to 6 for the definition of vitamin D deficiency but use ref 18 for the definition of vitamin D deficiency in this study. Why can't they use only reference 6 in the introduction and for the definition in this study given that it is the most recently published article? Can you specify whether you assessed the human form of 25-OH-D (25-OH-D3) throughout the manuscript?

Response: We thank the editor for your careful reading and constructive suggestions. We realize that the previous version of the manuscript cited multiple references regarding the deficiency of vitamin D level, which induced redundancy. We agree that ref 6 is the latest and authoritative source for defining vitamin D status. Therefore, we have unified citation of ref 6 in both introduction and methods sections, meanwhile remove ref 4-5 and 18 from our revised manuscript.

Regarding the detected form of 25-OH-D, we supplemented the following sentences in the Methods to improve the clarity of our manuscript: “From cycle 2007-2008 and beyond, liquid chromatography-tandem mass spectrometry (LC-MS/MS) was used to measure serum 25(OH)D concentrations, including 25(OH)D3, epi-25(OH)D3, and 25(OH)D2 in human serum. We calculated the 25(OH)D levels by adding 25(OH)D3 and 25(OH)D2 levels.” (marked in blue, Page 5).

7. After reading your introduction, I still do not well understand the rationale for conducting this study given that as well mentioned in reference 6 of the current manuscript, it is largely known that older individuals generally have lower vitamin D levels than younger populations. Do you want to asses whether that knowledge based on social age could hold even when using biological aging metrics (taking into consideration the discrepancy between social and biological ages)? The aim and rationale for conducting the study should be unambiguous in the last paragraph of the introduction. Still regarding the introduction, it is important to begin to be more cautious and specific when making claims about the involvement of vitamin D in human disease based on epidemiological data (see Vitamin D and human healrh: evidence from Mendelian randomization studies. Eur J Epidemiol. 2024;39: 467-490 and Schooling MC and Jones HE. Clarifying questions about "risk factors": predictors versus explanation. Epidemiology in Themes. 2018; 15:10). From my perspective, in a study whereby you seemingly assess the epidemiological link between vitamin D and physiological aging, the introduction should (in a brief overview form: introduction of no more than 1.5 pages) highlight how the human form of vitamin D evolves throughout the age spectrum based on the most certain available peer-reviewed evidence, as well as what is known and what is unknown about the epidemiological association between the human form of vitamin D and physiological aging, before stating the rationale for conducting the study (gaps in knowledge you would like to fill) and the study's aim.

Response: Thank you for your feedback regarding the introduction section of our manuscript. We have rewritten the introduction section to make it clear and concise in the revised manuscript. Briefly, the aim of current study is to investigate the relationship between vitamin D and physiological aging. We have read the articles mentioned above and rewrite the introduction of vitamin D in human disease based on epidemiological data. The length of introduction has been limited to less than 2 pages, a little longer than 1.5 pages. We have done our best to improve the induction section and hope this improvement could meet your requirement. Thank you very much.

8. Finally, language editing is also required.

Response: Thank you for your valuable feedback and suggestions for improvement. I acknowledge the need for language editing to enhance the clarity, precision, and professionalism of the manuscript. I will thoroughly review and ask help from professional language improvement to ensure that the text is clear and effectively conveys the intended meaning.

Reviewer #1:

Feng et al conduct this research in a rare direction while most of researchers will focus on the association between 25(OH)D and disease. Here are some issues should be concerned.

1. In the abstract, there is no statistical results. The conclusion that VD deficiency promotes aging is not proper since cross sectional study only remind the possible association between exposure and outcome instead of causuality.

Response: Thank you for your thoughtful feedback. We have revised the abstract to include key statistical results to better inform readers about the study's findings.

Additionally, we acknowledge your point regarding the limitations of cross-sectional studies in establishing causality. You are correct that such studies can only identify associations, and we will revise the language in the abstract and manuscript to reflect this appropriately. Specifically, we will modify the conclusion to state that vitamin D deficiency "may be associated with" or "shows a potential link to" biological aging, rather than implying causation.

2. The introduction section is a little long, in which 3-4 paragraphes can be better.

Response: Thank you for your suggestion. We acknowledge that the current introduction may be too lengthy and appreciate your suggestion to condense it into 3-4 focused paragraphs.

3. In methods section, data including flow chart should be added, and please learn some from the below articles:

a. Possible non-linear relation between prostate specific antigen and vitamin D: a machine learning study based on cross-section data

b. Association between urinary metals and prostate-specific antigen in aging population with depression: a cross-sectional study

Response: We thank the reviewer for your constructive suggestion. We have supplemented a flowchart that outline the participant selection process as Figure 1 in revised manuscript.

4. In the discussion section, this study should declare the mechanism in the increase of 25(OH)D acclerate aging.

Response: Thank you for your suggestion. We have rewritten the discussion section, providing the mechanism how the increase of 25(OH)D accelerate aging. “These detrimental effects may stem from the toxicity of excessive vitamin D. When vitamin D intake is too high, serum levels of its metabolites (25(OH)D and 24,25(OH)₂D) rise and surpass the binding capacity of vitamin D binding protein. This leads to an increase in the free active metabolite, 1,25(OH)₂D. This active form reaches the vitamin D receptor (VDR) in the nucleus, promoting gene expression that enhances calcium absorption and bone mobilization. Consequently, this can result in hypercalcemia or hyperphosphatemia, which in turn suppresses initial parathyroid hormone (iPTH) levels. These disrupting balance of calcium and hormone may lead to conditions such as cardiovascular disease,

---

## [Decision Letter · Decision Letter 1]

7 Apr 2025

Dear Dr. Wang,

Thank you for submitting your manuscript to PLOS ONE. After careful consideration, we feel that it has merit but does not fully meet PLOS ONE’s publication criteria as it currently stands. Therefore, we invite you to submit a revised version of the manuscript that addresses the points raised during the review process.

We look forward to receiving your revised manuscript.

Kind regards,

Mickael Essouma, M. D.

Academic Editor

PLOS ONE

Additional Editor Comments:

The authors have improved the manuscript. However, as mentioned by the reviewers, there is a a room for further improvement.

Suggestions about how to address the comments of Reviewer 5:

1. The terminologies "Physiological age" and "Biological age". Although the terminology "Physiological age" is known in the health literature to be based on some bodily functions as described in the article by Nakamura et al. Further evaluation of physical fitness age versus physiological age in women. Eur J Appl Physiol (1998) 78: 195-200 for example, the authors should indeed replace the terminology "Physiological age" with "biological age" where necessary because DNAm PhenoAge, the aging metric used in this study, is a composite of biomarkers, not bodily functions.

2. The race issue. The authors should stick to the racial stratification used in the 2007, 2008, 2009, 2010, 2011, 2012, 2013, 2014, 2015 and 2016 NHANES surveys. Indeed, this is not the first report on 25(OH)D assessment in a multiracial NHANES population as shown by the article Daraghmeh et al. Evidence for the vitamin D hypothesis: The NHANES III extended mortality follow-up. Atherosclerosis 255 (2016) 96e101. The lack of data segregation by race could be discussed as a common study limitation of NHANES surveys (https://pubmed.ncbi.nlm.nih.gov/?term=Vitamin+D+AND+NHANES) in the absence of race-specific cut-off values for vitamin D levels as for other biological biomarkers such as the creatinine-based estimated glomerular filtration rate, and when it is unclear whether 25-hydroxyvitamin serum levels ≥ 20 ng/mL are optimal across all racial groups in the USA: see The Journal of Clinical Endocrinology & Metabolism, Volume 109, Issue 8, August 2024, Pages 1948–1954, https://doi.org/10.1210/clinem/dgae322.

3. The cut-off value of 25-hydroxyvitamin D for vitamin D deficiency. Given that the reviewer did not provide a reference to support the value of 25 nmol/L proposed, I recommend the authors stick to 20 ng/mL (50 nmol/L) as the cut-off value to define optimum and low levels of 25-hydroxyvitamin D in this study: see J Clin Endocrinol Metab. 2024 Jul 12;109(8):1907-1947. doi: 10.1210/clinem/dgae290. and The Journal of Clinical Endocrinology & Metabolism, Volume 109, Issue 8, August 2024, Pages 1948–1954, https://doi.org/10.1210/clinem/dgae322. However, the authors need to implement the reviewer's comment in their discussion of study limitations. In the same vein, the lack of genetic data on isoform of Vitamin D binding protein (VBD) should be discussed as a study limitation because VBD is allegedly the main determinant of the unbound 25-hydroxyvitamin D which is postulated to be the active compound of 25(OH)D (Bikle and Schwatrz. Vitamin D Binding Protein, Total and Free Vitamin D Levels in Different Physiological and Pathophysiological Conditions. Front Endocrinol (Lausanne). 2019 May 28:10:317. doi: 10.3389/fendo.2019.00317.).

4. Contemporary evidence about vitamin D association with disease. As mentioned in the first round of review and further highlighted by Reviewer 5, the authors need to exercise caution when stating that vitamin D serum concentrations are associated with diseases and that vitamin D supplementation is beneficial in disease prevention. For example, see the articles by Fang A, Zhao Y, Yang P, Zhang X, and Giovannucci EL. Vitamin D and human health: evidence from Mendelian randomization studies. Eur J Epidemiol. 2024;39(5):467-90. Epub 2024/01/12. doi: 10.1007/s10654-023-01075-4. PubMed PMID: 38214845, Sun et al. Association between vitamin D level and risk of type 2 diabetes: a systematic review of Mendelian Randomization studies. Crit Rev Food Sci Nutr. 2025 Mar 12:1-10. doi: 10.1080/10408398.2025.2466758, J Clin Endocrinol Metab. 2024 Jul 12;109(8):1907-1947. doi: 10.1210/clinem/dgae290. and The Journal of Clinical Endocrinology & Metabolism, Volume 109, Issue 8, August 2024, Pages 1948–1954, https://doi.org/10.1210/clinem/dgae322.

5. The main source of vitamin D in human beings. I agree with Reviewer 5 that food intake may be the greatest source of vitamin D in White people living in the Transatlantic region where the sunny period is relatively limited. However, textbooks report vitamin D production by the skin as the main source of vitamin D in human beings, with food intake contributing to only up to 10% of the human body's store of vitamin D. The limited epidemiologic data on the source of vitamin D in the other ancestral groups in the transatlantic region and in tropical regions (for example, see the articles by Vearing, R.M., Hart, K.H., Darling, A.L. et al. Global Perspective of the Vitamin D Status of African-Caribbean Populations: A Systematic Review and Meta-analysis. Eur J Clin Nutr 76, 516–526 (2022). https://doi.org/10.1038/s41430-021-00980-9 and Lancet Glob Health. 2020 Jan;8(1):e134-e142. doi: 10.1016/S2214-109X(19)30457-7. Epub 2019 Nov 27) does not allow the verification of this textbook knowledge in these other ancestral groups. However, vitamin D production by the skin could be the predominant source of vitamin D in the non-White population (at least those with a melanoderm skin), given their much higher exposure to sun-derived UVB radiations throughout the year with co-existent undernutrition when compared with White populations.

I have also appended additional comments in the attached pdf.

Reviewers' comments:

Reviewer's Responses to Questions

**Comments to the Author**

Reviewer #1: All comments have been addressed

Reviewer #2: All comments have been addressed

Reviewer #3: All comments have been addressed

Reviewer #4: (No Response)

Reviewer #5: (No Response)

Reviewer #6: (No Response)

2. Is the manuscript technically sound, and do the data support the conclusions?

Reviewer #1: Yes

Reviewer #2: Yes

Reviewer #3: Yes

Reviewer #4: Yes

Reviewer #5: No

Reviewer #6: Yes

3. Has the statistical analysis been performed appropriately and rigorously?

Reviewer #1: Yes

Reviewer #2: Yes

Reviewer #3: Yes

Reviewer #4: Yes

Reviewer #5: No

Reviewer #6: I Don't Know

4. Have the authors made all data underlying the findings in their manuscript fully available?

Reviewer #1: Yes

Reviewer #2: Yes

Reviewer #3: Yes

Reviewer #4: Yes

Reviewer #5: Yes

Reviewer #6: Yes

5. Is the manuscript presented in an intelligible fashion and written in standard English?

Reviewer #1: Yes

Reviewer #2: Yes

Reviewer #3: Yes

Reviewer #4: Yes

Reviewer #5: Yes

Reviewer #6: Yes

Reviewer #1: All revisions were made and there is no further comment. The revised manuscript meet the journal publish standard.

Reviewer #2: There is no comment. In my opinion, the present version of the manuscript is ready for publication. However, please check the plagiarism of the whole text and be sure that the similarity index was less than 30 %.

Reviewer #3: Authors addressed most of the previous comments from reviewers. The manuscript reads better in terms of flow and clarity. However further language editing is still required in many sections throughout the manuscript.

1. In the abstract, authors to reflect the U shape effect observed among the males; improve clarity of sentences

2. In methodology, citing previous publications, authors to briefly elaborate on each biomarkers/model. At present, there's gap of information for these modelling.

3. Is there methodology established to determine physiological aging? if there is, how does the PhenoAge algorithm fare when compared? This would further establish validity of methodology.

4. In discussion and results, there's inconsistency in reporting ie ie dose-dependent relationship in females

5. In discussion, further strengthening of write up required. At present, it is unclear if low vit d level drives/associated with physiological ageing or is an impact of ageing process.

Reviewer #4: Feng et al. investigated the association between vitamin D deficiency and physiological aging. While the authors have made considerable efforts to address previous reviewers’ comments, I believe the manuscript still requires further revision. Below are my comments:

• The Introduction remains somewhat lengthy. Additionally, please clarify that endogenous vitamin D production in humans and animals yields vitamin D3, not D2 (see line 43).

• Appropriate references should be added for the activation steps of vitamin D, including both the 25-hydroxylation and 1α-hydroxylation processes.

• Line 47: Please correct the terminology to "1α,25-dihydroxyvitamin D."

• Line 49: Choose either reference 1 or 2, citing both is unnecessary.

* Throug

• The criteria for defining vitamin D status are based solely on the Endocrine Society guidelines. The Institute of Medicine (IOM) guidelines, which differ substantially and are widely used in clinical settings, should also be mentioned. See ref below:

A. Catharine Ross, JoAnn E. Manson, Steven A. Abrams, John F. Aloia, Patsy M. Brannon, Steven K. Clinton, Ramon A. Durazo-Arvizu, J. Christopher Gallagher, Richard L. Gallo, Glenville Jones, Christopher S. Kovacs, Susan T. Mayne, Clifford J. Rosen, Sue A. Shapses, The 2011 Report on Dietary Reference Intakes for Calcium and Vitamin D from the Institute of Medicine: What Clinicians Need to Know, The Journal of Clinical Endocrinology & Metabolism, Volume 96, Issue 1, 1 January 2011, Pages 53–58, https://doi.org/10.1210/jc.2010-2704

• The rationale for the study remains unclear. The authors should clarify their aims more.

• The sentence: “However, the comprehensive picture of the association between vitamin D levels and physiological aging remains to be fully elucidated, particularly in the general population without vitamin D deficiency or insufficiency” is vague. Please revise to clarify what is meant by “general population without deficiency or insufficiency.”

• Although several covariates were adjusted for, the possibility of residual confounding, particularly from factors such as sun exposure and vitamin D supplementation, should be acknowledged in the discussion.

• Recent research has provided new insights into vitamin D metabolism and its regulation by metabolic state:

Aatsinki and Elkhwanky et al. (2019) demonstrated that fasting and diabetes activate at least two transcriptional regulators i.e., PGC-1α/ERRα and the glucocorticoid receptor (GR), which repress CYP2R1 and thereby reduce bioactivation of vitamin D. These findings question the long-held assumption that vitamin D levels solely reflect intake and sun exposure.

Elkhwanky et al. (2020) showed that obesity suppresses CYP2R1, the enzyme responsible for vitamin D 25-hydroxylation, in both hepatic and extrahepatic tissues.

These two references below need to be cited and discussed in the discussion section:

Aatsinki SM*, Elkhwanky MS*, Kummu O, Karpale M, Buler M, Viitala P, Rinne V, Mutikainen M, Tavi P, Franko A, Wiesner RJ, Chambers KT, Finck BN & Hakkola J

(2019) Fasting-Induced Transcription Factors Repress Vitamin D Bioactivation, a

Mechanism for Vitamin D Deficiency in Diabetes. Diabetes 68(5): 918-931.

https://doi.org/10.2337/db18-1050.

Elkhwanky MS, Kummu O, Piltonen TT, Laru J, Morin-Papunen L, Mutikainen M,

Tavi P & Hakkola J (2020) Obesity Represses CYP2R1, the Vitamin D

25-Hydroxylase, in the Liver and Extrahepatic Tissues. JBMR PLUS https://doi.org/10.1002/jbm4.10397.

These findings collectively challenge the traditional view that 25-hydroxylation is a stable, unregulated process, emphasizing instead the influence of metabolic state on vitamin D status. They also raise important questions about the suitability of 25(OH)D as a reliable biomarker in populations with metabolic disorders including diabetes. While 25(OH)D remains the most accepted marker of vitamin D status, these studies should be cited and discussed, particularly in relation to the study’s diabetic subgroup.

• Line 261: Please remove either “diseases” or “disorders” to avoid redundancy.

• Lines 313–316: A reference on vitamin D toxicity should be included to support the statement.

• Some phrases still imply causality (e.g., "vitamin D deficiency promotes aging"). Since this is a cross-sectional study, causal language must be avoided.

• The suggestion that high vitamin D levels may accelerate aging in men appears speculative and requires either stronger supporting evidence or a more cautious tone.

• English language revision is recommended.

In summary, although this is a timely and relevant study with notable improvements, several key areas including the clarity of the rationale, the discussion of mechanistic insights, and the interpretation of results, still require revision. Therefore, I recommend a major revision.

Reviewer #5: This is a study on chronological and biological aging and 25(OH-D-vitamin based on data from NHANES.

Major issues:

The first problem is how the authors treat age. Chronological age is simple, calculated biological age is also relevant. Phenoage is not used as defined by omitting some of the variables defining biological age.

The authors also define a concept called physiological age, which is also referenced in the title. This is a major problem.

Page 16 line “We further calculated physiological aging as the difference between PhenoAge and chronological age.“

This is a confusing terminology, not only for the readers, but the authors also have difficulties with this:

In the abstract line 21 “Biological age showed a negative correlation with serum 25(OH)D levels (β=-0.04; 95% CI: -0.08-0.00, p=0.039)”

In the results page 18 line 211: “physiological aging was found to be negatively associated with serum 25(OH)D levels (β=-0.04; 95% CI: -0.08-0.00).”

Omit the term' physiological age' and name it for what it really is.

Write instead: The difference between Phenoage and age was negatively correlated serum 25(OH)D levels …

Major cohort problem

In NHANES, as in the rest of the US, race has been defined in different ways during the years. For example if one of the parents are of different races and one is white, the child is classified according to the race of the other parent. When neither parent is white the child is classified according to the fathers race. Other ways of defining race are in use, such as self-stated, decision by the doctor, or even name analysis. This is a major problem when discussing vitamin D in relation in a mixed cohort like the one used by the author. How big percentage would define race?

As has been thoroughly discussed about the inherent errors caused by race-adjusted equations for estimating glomerular filtration rate based on creatinine, this also applies to a study on vitamin D.

The main determinant for 25(OH)-D-vitamin in serum/plasma is a genetic variant of vitamin D-binding protein = GC-globulin. There are three isoforms, Gc1, Gc1-2, and Gc2-2 isoforms. The genetic distribution of these isoforms is highly dependent on ethic origin. The three isoforms come with different 25(OH)-D vitamin levels, although this does not affect bioavailable vitamin D not measured in this and many other studies. Thus, there would be a need for different cut-offs for deficiency, depending on the isoforms. For example, vitamin D is associated with colorectal cancer only in one of these isoforms, and the isoforms modify the risk for bladder cancer.

Thus, it can be concluded that the NHANES study is not well suited for this kind of study applying knowledge from the last ten years. Perhaps if using whites only, but it is not biologically accurate to use a composite population like the authors do.

The authors state incorrectly that vitamin D is associated with cardiovascular disease and osteoporosis. This has been proven incorrect over the last decade. There remains an association with some inflammatory diseases such as multiple sclerosis and rheumatoid arthritis.

There is no “growing body of evidence” to support a statement of the potential of vitamin D supplementation to promote overall health.

Minor issues:

Vitamin D is more common in southern compared to northern Europe (PMID: 34933700), as vitamin D is probably more dependent on food intake compared to sun exposure.

The cut-off for deficiency at 50 nmol per liter is not solid, it is a consensus not anchored in facts. A more likely cut-off would be 25 nmol/L.

It is, however, interesting that higher levels are less beneficial in terms of biological age. This is not in line with the recommendations from organisations associated with the industry for supplementing vitamin D, aiming to increase vitamin D levels in the population.

Reviewer #6: In this manuscript by Feng et al, the authors aimed to explore the link between blood vitamin D levels and physiological aging, using data from over 13,000 U.S. adults in NHANES (2007–2016). The physiological aging was estimated using the PhenoAge model. The results revealed a nonlinear relationship: lower vitamin D levels were generally associated with accelerated aging, especially in females, while excessively high levels were linked to faster aging in males. These patterns suggest that both deficiency and excess of vitamin D may impact aging differently by gender. This revised manuscript has addressed most of the concerns raised by the Editor and Reviewer. However, there are a number of points that still need to be clarified. Specific comments are as follows:

1) The authors mentioned in the methods that CRP numbers were missing in some NHANES data and that PhenoAge (excluding CRP) was used. It would be helpful to briefly comment on how omitting CRP might affect comparability with studies using the standard PhenoAge metric.

2) In Line 220, the authors cite a p-value (p = 0.076) to describe the absence of dose-dependent trends in both the whole population and females. It is suggested to clarify this sentence to correctly distinguish between the groups with the right numbers.

3) In the Discussion section (line 323), the authors state that serum 25(OH)D levels above 125 nmol/L in female participants were linked to accelerated physiological aging. However, this interpretation does not appear to be supported by the data in Table 3. Revising this statement to more accurately reflect the results is recommended.

**Do you want your identity to be public for this peer review?** For information about this choice, including consent withdrawal, please see our Privacy Policy

Reviewer #1: No

Reviewer #2: No

Reviewer #3: No

Reviewer #4: No

Reviewer #5: No

Reviewer #6: No

---

## [Author Response · Author response to Decision Letter 2]

14 May 2025

Dear editor and reviewers:

Thank you very much for your careful review and constructive suggestions with regard to our manuscript. Those comments are helpful for us to revise and improve our paper. We have studied comments carefully and tried our best to revise and improve the manuscript and made great changes in the manuscript according to the good comments. We earnestly appreciate for the warm work from you, and hope that the corrections will meet with approval. Please feel free to contact us with any questions and we are looking forward to your consideration. The main corrections in the paper and the responds to your comments are as following:

Additional Editor Comments:

The authors have improved the manuscript. However, as mentioned by the reviewers, there is a room for further improvement.

Suggestions about how to address the comments of Reviewer 5:

1. The terminologies "Physiological age" and "Biological age". Although the terminology "Physiological age" is known in the health literature to be based on some bodily functions as described in the article by Nakamura et al. Further evaluation of physical fitness age versus physiological age in women. Eur J Appl Physiol (1998) 78: 195-200 for example, the authors should indeed replace the terminology "Physiological age" with "biological age" where necessary because DNAm PhenoAge, the aging metric used in this study, is a composite of biomarkers, not bodily functions.

Response: we have replaced the "Physiological age" with "biological age". Thank you for this suggestion.

2. The race issue. The authors should stick to the racial stratification used in the 2007, 2008, 2009, 2010, 2011, 2012, 2013, 2014, 2015 and 2016 NHANES surveys. Indeed, this is not the first report on 25(OH)D assessment in a multiracial NHANES population as shown by the article Daraghmeh et al. Evidence for the vitamin D hypothesis: The NHANES III extended mortality follow-up. Atherosclerosis 255 (2016) 96e101. The lack of data segregation by race could be discussed as a common study limitation of NHANES surveys (https://pubmed.ncbi.nlm.nih.gov/?term=Vitamin+D+AND+NHANES) in the absence of race-specific cut-off values for vitamin D levels as for other biological biomarkers such as the creatinine-based estimated glomerular filtration rate, and when it is unclear whether 25-hydroxyvitamin serum levels ≥ 20 ng/mL are optimal across all racial groups in the USA: see The Journal of Clinical Endocrinology & Metabolism, Volume 109, Issue 8, August 2024, Pages 1948–1954, https://doi.org/10.1210/clinem/dgae322IF: 5.0 Q1 .

Response: the discussion of race has been added in the revised manuscript. thanks for your suggestion. (Page14, line 371-381)

3. The cut-off value of 25-hydroxyvitamin D for vitamin D deficiency. Given that the reviewer did not provide a reference to support the value of 25 nmol/L proposed, I recommend the authors stick to 20 ng/mL (50 nmol/L) as the cut-off value to define optimum and low levels of 25-hydroxyvitamin D in this study: see J Clin Endocrinol Metab. 2024 Jul 12;109(8):1907-1947. doi: 10.1210/clinem/dgae290IF: 5.0 Q1 . and The Journal of Clinical Endocrinology & Metabolism, Volume 109, Issue 8, August 2024, Pages 1948–1954, https://doi.org/10.1210/clinem/dgae322IF: 5.0 Q1 . However, the authors need to implement the reviewer's comment in their discussion of study limitations. In the same vein, the lack of genetic data on isoform of Vitamin D binding protein (VBD) should be discussed as a study limitation because VBD is allegedly the main determinant of the unbound 25-hydroxyvitamin D which is postulated to be the active compound of 25(OH)D (Bikle and Schwatrz. Vitamin D Binding Protein, Total and Free Vitamin D Levels in Different Physiological and Pathophysiological Conditions. Front Endocrinol (Lausanne). 2019 May 28:10:317. doi: 10.3389/fendo.2019.00317.).

Response: we have added a paragraph discussing the cut-off value of 25-hydroxyvitamin D for vitamin D deficiency. (Page 13, line 332-352)

“Regarding the threshold of vitamin D deficiency, it remains an ongoing controversy. The Scientific Advisory Committee on Nutrition (2016) recommended threshold at 25 nmol/L among UK population [46], whereas the IOM (2011) suggested thresholds at 30 nmol/L based on evidence in U.S. and Canadian populations [20]. Meanwhile, the European Food Safety Authority 2016 and the Endocrine Society guideline 2011 proposed higher level of 50 nmol/L as deficiency cutoff [47, 48]. The discrepancy in cutoff values stems from multiple factors, for example, different regional ultraviolet B exposure patterns, dietary practice and population-specific biological variability. Furthermore, guideline variations also reflect different focus on health outcomes, with some prioritizing musculoskeletal health [49]. In 2024, the Endocrine Society guideline retracted their previous recommendation of threshold at 50 nmol/L due to limited clinical trial evidence [7]. It also highlighted the impracticality of universal thresholds due to heterogeneity in health outcomes and population characteristics. Hence, the interpretation of 25(OH)D level threshold determination in our study should be with caution, especially when it derived from observational study in an ethnically diverse population. Nevertheless, the trend of benefits attenuation we observed is worth noting, suggesting excessive level of serum 25(OH)D may not be beneficial regarding biological aging process.”

4. Contemporary evidence about vitamin D association with disease. As mentioned in the first round of review and further highlighted by Reviewer 5, the authors need to exercise caution when stating that vitamin D serum concentrations are associated with diseases and that vitamin D supplementation is beneficial in disease prevention. For example, see the articles by Fang A, Zhao Y, Yang P, Zhang X, and Giovannucci EL. Vitamin D and human health: evidence from Mendelian randomization studies. Eur J Epidemiol. 2024;39(5):467-90. Epub 2024/01/12. doi: 10.1007/s10654-023-01075-4IF: 7.7 Q1 . PubMed PMID: 38214845IF: 7.7 Q1 , Sun et al. Association between vitamin D level and risk of type 2 diabetes: a systematic review of Mendelian Randomization studies. Crit Rev Food Sci Nutr. 2025 Mar 12:1-10. doi: 10.1080/10408398.2025.2466758IF: 7.3 Q1 , J Clin Endocrinol Metab. 2024 Jul 12;109(8):1907-1947. doi: 10.1210/clinem/dgae290IF: 5.0 Q1 . and The Journal of Clinical Endocrinology & Metabolism, Volume 109, Issue 8, August 2024, Pages 1948–1954, https://doi.org/10.1210/clinem/dgae322IF: 5.0 Q1 .

Response: we have checked the expression of our result and conclusion to avoid expressing causal relationships. Our study is only a cross-sectional epidemiological research.

5. The main source of vitamin D in human beings. I agree with Reviewer 5 that food intake may be the greatest source of vitamin D in White people living in the Transatlantic region where the sunny period is relatively limited. However, textbooks report vitamin D production by the skin as the main source of vitamin D in human beings, with food intake contributing to only up to 10% of the human body's store of vitamin D. The limited epidemiologic data on the source of vitamin D in the other ancestral groups in the transatlantic region and in tropical regions (for example, see the articles by Vearing, R.M., Hart, K.H., Darling, A.L. et al. Global Perspective of the Vitamin D Status of African-Caribbean Populations: A Systematic Review and Meta-analysis. Eur J Clin Nutr 76, 516–526 (2022). https://doi.org/10.1038/s41430-021-00980-9IF: 3.6 Q2 and Lancet Glob Health. 2020 Jan;8(1):e134-e142. doi: 10.1016/S2214-109X(19)30457-7IF: 19.9 Q1 . Epub 2019 Nov 27) does not allow the verification of this textbook knowledge in these other ancestral groups. However, vitamin D production by the skin could be the predominant source of vitamin D in the non-White population (at least those with a melanoderm skin), given their much higher exposure to sun-derived UVB radiations throughout the year with co-existent undernutrition when compared with White populations.I have also appended additional comments in the attached pdf.

Response: sun exposure and food intake are main sources of vitamin D. We totally agree this comment from reviewer and editor. The discussion has been added in the section of limitation. We have carefully read the additional comments added from editor. thank you so much for helping us improving our manuscript.

Reviewers' comments:

Reviewer #1: All revisions were made and there is no further comment. The revised manuscript meet the journal publish standard.

Response: Thank you so much.

Reviewer #2: There is no comment. In my opinion, the present version of the manuscript is ready for publication. However, please check the plagiarism of the whole text and be sure that the similarity index was less than 30 %.

Response: Thank you so much. We have checked the plagiarism of manuscript and it meets the journal standards.

Reviewer #3: Authors addressed most of the previous comments from reviewers. The manuscript reads better in terms of flow and clarity. However further language editing is still required in many sections throughout the manuscript.

1. In the abstract, authors to reflect the U shape effect observed among the males; improve clarity of sentences

Response: we are sorry for this language problem. In the revised abstract, we clearly state the U-shaped effect, provides context for its significance, and ensures each sentence is concise and informative.

2. In methodology, citing previous publications, authors to briefly elaborate on each biomarkers/model. At present, there's gap of information for these modelling.

Response: We thank the reviewer for your constructive comments. To fully elucidate the rationale for selecting analysis models, we have supplemented the following sentences in the Methods section: “To explore the non-linear relationship between serum 25(OH)D and physiological aging, generalized additive model (GAM) was constructed. The GAMs extend traditional linear regression by replacing linear terms with smooth functions, allowing flexible modeling of complex dose-response relationships without assuming a priori parametric forms. The threshold effect was further estimated using segmented regression model. This method identifies inflection points where the slope of the association between 25(OH)D and PhenoAgeAccel changes significantly. The optimal breakpoint was determined via likelihood ratio tests, comparing the goodness-of-fit between the non-segmented linear model and the segmented regression model.” (Page 7, line 180-189).

The following references were supplemented in the References List:

28 Hastie TJ. Generalized additive models. Statistical models in S. 2017:249-307.

29 Muggeo VM. Estimating regression models with unknown break-points. Stat Med. 2003;22(19):3055-71. Epub 2003/09/16. doi: 10.1002/sim.1545. PubMed PMID: 12973787.

3. Is there methodology established to determine physiological aging? if there is, how does the PhenoAge algorithm fare when compared? This would further establish validity of methodology.

Response: We thank the reviewer for your constructive comments. In agree with the editor’s suggestion, the terminology has been revised from "physiological aging" to "biological aging" throughout the manuscript, as this better aligns with the definition of PhenoAge—a composite metric integrating 10 clinical biomarkers that collectively reflect systemic inflammatory, metabolic, and hematopoietic dysfunction. There are several established methods for quantification of biological aging, including molecular clocks (e.g., DNA methylation clocks), functional decline indices (e.g., frailty index) and composite biomarker metrics (e.g., PhenoAge). The PhenoAge algorithm was constructed based on NHANES III (1988–1994), and was validated using data form NHANES IV (1999–2010). Hence, in NHANES dataset, the PhenoAge method showed good data compatibility and epidemiological validity. In the Methods section, we have supplemented the following sentences to elucidate the rationale for choosing PhenoAge as biological aging indicator: “In this study, we utilized PhenoAge as the indicator of biological aging, considering its high robustness and feasibility in NHANES data as a reliable predictor of mortality and age-related disease risks. The PhenoAge algorithm is constructed using NHANES III (1988–1994) by elastic-net regression, encompasses 10 markers: albumin, creatinine, glucose, white blood cell count, lymphocyte percent, red cell distribution width, mean red cell volume, C-reactive protein, alkaline phosphatase and chronological age [21]. PhenoAge was both developed and validated within NHANES populations. Meanwhile, all constituent biomarkers are routinely measured in NHANES with standardized protocols, ensuring reproducibility across study cycles.” (Page 5, line 118-140).

4. In discussion and results, there's inconsistency in reporting ie ie dose-dependent relationship in females

Response: we have improved this description in the section of discussion and results. Thank you for your suggestion.

5. In discussion, further strengthening of write up required. At present, it is unclear if low vit d level drives/associated with physiological ageing or is an impact of ageing process.

Response: Numerous evidences have found that vitamin D may play a role in modulating key biological pathways associated with chronological age. Vitamin D is known to influence cellular senescence, oxidative stress, and inflammation, which are central to the aging process. Furthermore, experimental studies have demonstrated that vitamin D supplementation can mitigate oxidative damage and improve immune function in elder, suggesting its potential role in slowing age-related physiological decline. Thus, we did not provide enough information about this aspect. In the revised manuscript, we have added this in the discussion. Moreover, we have asked professional help to improve our language. Thank you so much for your review.

Reviewer #4: Feng et al. investigated the association between vitamin D deficiency and physiological aging. While the authors have made considerable efforts to address previous reviewers’ comments, I believe the manuscript still requires further revision. Below are my comments:

1. The Introduction remains somewhat lengthy. Additionally, please clarify that endogenous vitamin D production in humans and animals yields vitamin D3, not D2 (see line 43).

Response: thank you for your suggestion. The revised introduction is more streamlined, removing unnecessary details while maintaining essential information.

Vitamin D exists in two primary forms: vitamin D3 (cholecalciferol) and vitamin D2 (ergocalciferol). D3 is produced from 7-dehydrocholesterol (7-DHC) through a two-step process in which the B ring is broken by UV light (spectrum 280–320 UVB) radiation from the sun, forming pre-D3 that isomerizes to D3 in a thermo-sensitive but noncatalytic process. Both UVB intensity and skin pigmentation level contribute to the rate of D3 formation. Vitamin D3 is synthesized endogenously in the skin of humans and animals upon sunlight exposure, while vitamin D2 is primarily derived from dietary sources.

2. Appropriate references should be added for the activation steps of vitamin D, including both the 25-hydroxylation and 1α-hydroxylation processes.

Response: we have checked the reference and added appropriate one in the section of introduction.

3. Line 47: Please correct the terminology to "1α,25-dihydroxyvitamin D."

Response: Sorry for this mistake and we have corrected it. Thank you so much.

4. Line 49: Choose either reference 1 or 2, citing both is unnecessary.

Response: In the revised manuscript, reference 2 was removed.

5. The criteria for defining vitamin D status are based solely on the Endocrine Society guidelines. The Institute of Medicine (IOM) guidelines, which differ substantially and are widely used in clinical settings, should also be mentioned. See ref below:

A. Catharine Ross, JoAnn E. Manson, Steven A. Abrams, John F. Aloia, Patsy M. Brannon, Steven K. Clinton, Ramon A. Durazo-Arvizu, J. Christopher Gallagher, Richard L. Gallo, Glenville Jones, Christopher

---

## [Decision Letter · Decision Letter 2]

1 Jun 2025

Dear Dr. Wang,

Thank you for submitting your manuscript to PLOS ONE. After careful consideration, we feel that it has merit but does not fully meet PLOS ONE’s publication criteria as it currently stands. Therefore, we invite you to submit a revised version of the manuscript that addresses the points raised during the review process.

We look forward to receiving your revised manuscript.

Kind regards,

Mickael Essouma, M. D.

Academic Editor

PLOS ONE

Additional Editor Comments:

The reviewers have recommended the acceptance of the current manuscript for publication. However, not only did the authors partly address my comments, but there are also multiple reporting issues that lead me to request a minor revision before the manuscript can be accepted for publication. 

1. Consider addressing the points 2, 3, 4 and 5 highlighted in the section "Additional Editor comments" of the Decision letter about manuscript R1 when discussing the study limitations in the discussion section.

2. The first major reporting issue is the lack of clarity about the metrics of biological aging studied.  Was it PhenoAge alone, PhenoAge and PhenoAgeAccel, or PhenoAgeAccel alone? This issue stems from the fact that the term PhenoAge is inconsistently used throughout the manuscript. In the same vein, it seems to me that developers of PhenoAge termed it "DNAm PhenoAge". So, even if many authors keep using the term "PhenoAge" rather than "DNAm PhenoAge", I would use the term "DNAm PhenoAge" and consequently also the term "DNAm PhenoAgeAccel" when talking about PHenoAge Acceleration throughout the manuscript. The choice of the term to use between "DNAm PhenoAge" and "PhenoAge" (and consequently also between "DNAm PhenoAgeAccel" and "PhenoAgeAccel") is up to you, the only requeirement in this regard is consistency with the term chosen.

3. A second major issue is that the authors do not use the PhenoAge classification (positive versus negative) when reporting on the association between 25(OH)D serum levels and PhenoAgeAccel throughout the manuscript, although they clearly mentioned that classification in the Materials and Methods section. Rather, they kept using the expression "reduced PhenoAgeAccel" which would need to be replaced by "negative PhenoAgeAccel" or perhaps "Decreasing PhenoAGeAccel". Therefore, consider implementing the recognised PhenoAgeAccel classification in the results and discussion sections of the manuscript.

4. A third major issue (linked to the other major study variable: 25(OH)D serum level) is that the authors sticked to the 2011 IOM classification of 25(OH)D serum levels which includes four vitamin D serum level categories (sufficiency, insufficiency, deficiency and equal to or above 75 nmol/L) meanwhile they were advised to drop that classification given that the Endocrine Society no longer endorses that vitamin D classification status. I get it that the authors have the right to choose the classification system that best suits their needs. However, when doing so, they need to acknowledge in the limitations statement of the discussion section that the classification system used is now outdated and few evidence if any, currently to supports that classification system.

5. In addition, 25(OH)D serum levels need to be expressed using either or both of the two units ng/mL and/or nmol/L, but the unit chosen should be consistently used throughout the manuscript. I advise against using the term "correlation" throughout the manuscript, given the lack of correlation analyses in the manuscript. More comments and suggested edits are available in the document PONE-D-24-25180_R2_Mickael Essouma comments and suggested edits.pdf attached to this decision letter.

Mickael Essouma

Reviewers' comments:

Reviewer's Responses to Questions

**Comments to the Author**

Reviewer #3: All comments have been addressed

Reviewer #4: All comments have been addressed

Reviewer #6: All comments have been addressed

2. Is the manuscript technically sound, and do the data support the conclusions?

Reviewer #3: Yes

Reviewer #4: Yes

Reviewer #6: Yes

3. Has the statistical analysis been performed appropriately and rigorously?

Reviewer #3: Yes

Reviewer #4: Yes

Reviewer #6: Yes

4. Have the authors made all data underlying the findings in their manuscript fully available?

Reviewer #3: Yes

Reviewer #4: Yes

Reviewer #6: Yes

5. Is the manuscript presented in an intelligible fashion and written in standard English?

Reviewer #3: Yes

Reviewer #4: Yes

Reviewer #6: Yes

Reviewer #3: Authors have addressed all the comments raised previously and there is no further comment. The revised manuscript is ready for publication.

Reviewer #4: I thank the authors for considering my comments when revising the paper. I have no further comments and I think the paper is now in a good shape for the publication. All the best!

Reviewer #6: (No Response)

**Do you want your identity to be public for this peer review?** For information about this choice, including consent withdrawal, please see our Privacy Policy

Reviewer #3: **Yes: ** Megan Chong Hueh Zan

Reviewer #4: No

Reviewer #6: No

---

## [Author Response · Author response to Decision Letter 3]

11 Jun 2025

Dear editor Mickael Essouma and reviewers

We sincerely thank the reviewers for their approval of our manuscript and the editor for the valuable and constructive suggestions, especially the meticulous word-for-word revisions. We have carefully studied all comments and incorporated them thoroughly into our revisions. We are very grateful for the editor's dedicated support and hope the revised manuscript is now satisfactory. Should you have any questions, please feel free to contact us. We appreciate your continued consideration.

1. Consider addressing the points 2, 3, 4 and 5 highlighted in the section "Additional Editor comments" of the Decision letter about manuscript R1 when discussing the study limitations in the discussion section.

Response The limitations have been revised in the manuscript, and the concerns raised in the "Additional Editor comments," including the vitamin D classification and race definition, have been addressed. Thank you so much.

2. The first major reporting issue is the lack of clarity about the metrics of biological aging studied. Was it PhenoAge alone, PhenoAge and PhenoAgeAccel, or PhenoAgeAccel alone? This issue stems from the fact that the term PhenoAge is inconsistently used throughout the manuscript. In the same vein, it seems to me that developers of PhenoAge termed it "DNAm PhenoAge". So, even if many authors keep using the term "PhenoAge" rather than "DNAm PhenoAge", I would use the term "DNAm PhenoAge" and consequently also the term "DNAm PhenoAgeAccel" when talking about PHenoAge Acceleration throughout the manuscript. The choice of the term to use between "DNAm PhenoAge" and "PhenoAge" (and consequently also between "DNAm PhenoAgeAccel" and "PhenoAgeAccel") is up to you, the only requeirement in this regard is consistency with the term chosen.

Response� We have reviewed the entire manuscript and made corrections to use either "PhenoAge" or "PhenoAgeAccel" accordingly.

3. A second major issue is that the authors do not use the PhenoAge classification (positive versus negative) when reporting on the association between 25(OH)D serum levels and PhenoAgeAccel throughout the manuscript, although they clearly mentioned that classification in the Materials and Methods section. Rather, they kept using the expression "reduced PhenoAgeAccel" which would need to be replaced by "negative PhenoAgeAccel" or perhaps "Decreasing PhenoAGeAccel". Therefore, consider implementing the recognised PhenoAgeAccel classification in the results and discussion sections of the manuscript.

Response�A positive value of PhenoAgeAccel indicates an accelerated biological aging process, while a negative value indicates a slower biological aging process. We have revised "reduced/increased PhenoAgeAccel" to "negative/positive PhenoAgeAccel".

4. A third major issue (linked to the other major study variable: 25(OH)D serum level) is that the authors sticked to the 2011 IOM classification of 25(OH)D serum levels which includes four vitamin D serum level categories (sufficiency, insufficiency, deficiency and equal to or above 75 nmol/L) meanwhile they were advised to drop that classification given that the Endocrine Society no longer endorses that vitamin D classification status. I get it that the authors have the right to choose the classification system that best suits their needs. However, when doing so, they need to acknowledge in the limitations statement of the discussion section that the classification system used is now outdated and few evidence if any, currently to supports that classification system.

Response� We have added this information in the section of limitation. “the classification of 25(OH)D serum levels used in our study is based on the 2011 IOM guidelines; however, more recent classifications, such as those from the Scientific Advisory Committee on Nutrition (2016), are considered more accurate. It would be preferable to analyze the association between vitamin D status and biological aging using these updated classifications”. Thank you very much.

5. In addition, 25(OH)D serum levels need to be expressed using either or both of the two units ng/mL and/or nmol/L, but the unit chosen should be consistently used throughout the manuscript. I advise against using the term "correlation" throughout the manuscript, given the lack of correlation analyses in the manuscript. More comments and suggested edits are available in the document PONE-D-24-25180_R2_Mickael Essouma comments and suggested edits.pdf attached to this decision letter.

Response�25(OH)D serum levels were expressed using nmol/L and we correct these through the whole manuscript.

In the revised manuscript, the term “correlation” has been replaced with “association”.

We have carefully reviewed all the comments in the document and accepted the suggested revisions. These suggestions have been very helpful in improving our manuscript. We are very grateful for the editor's support and revision. Thank you very much.

---

## [Editor Report · Decision Letter 3]

15 Jun 2025

Dear Dr. Wang,

Thank you for submitting your manuscript to PLOS ONE. After careful consideration, we feel that it has merit but does not fully meet PLOS ONE’s publication criteria as it currently stands. Therefore, we invite you to submit a revised version of the manuscript that addresses the points raised during the review process.

We look forward to receiving your revised manuscript.

Kind regards,

Mickael Essouma, M. D.

Academic Editor

PLOS ONE

Journal Requirements:

**Additional Editor Comments:**

The authors have continued to improve upon their manuscript, but need to polish the presentation of the materials and methods section and further edit the manuscript to improve the language quality.

Materials and Methods section.

Lines 159-164: consider addressing my request to provide references supporting the classification made for sleep duration, hypertension and diabetes.

As commented in the last round of review, you need to clearly specify that you conducted effect modification analyses for the association between 25(OH)D serum levels and PhenoAgeAccel in the statistical analysis sub-section before making the comment currently available in line 179. These references may be helpful in this respect: Edwarsd JR. Methods for Integrating Moderation and Mediation: A General Analytical Framework Using Moderated Path Analysis. Psychological Methods 2007, Vol. 12, No. 1, 1–22 and Knol et al. Recommendations for presenting analyses of effect modification and interaction. International Journal of Epidemiology 2012;41:514–520 doi:10.1093/ije/dyr218.

This statement appears in the abstract: "Generalized additive models further found 25(OH)D serum 25 levels were linked to negative PhenoAgeAccel at levels below 38.2 nmol/L (15.3 ng/mL) in males (β=-0.013; 95% CI: -0.025 to -0.002) and 62.5 nmol/L (25.0 ng/mL) in females (β=-0.007; 95% CI: -0.01 to -0.004,), revealing a U-shaped relationship observed in males but an L-shaped pattern in females. . However, 25(OH)D serum levels above 125 nmol/L were not associated with PhenoAgeAccel in females (β=-0.001; 95% CI: -0.005-0.002), while in males, elevated levels (>91.55 nmol/L) were associated with positive PhenoAgeAccel (β=0.005; 95% CI: 0.002-0.008).". As requested in the last round of review, consider specifying in the methods section of the main manuscript the reasons why you chose those cut-off values for serum 25-vitamin D while performing the corresponding statistical analyses.

Results section. As commented in the last round of review, consider rounding all the numbers to one decimal place when providing results throughout the manuscript (Table 1 and so on).

Comment appended to Table 1.

Discussion section: limitations statement. You also need to address the lack of data on vitamin D supplement among the confounders, the effect of unmeasured confounding and the reverse causation and residual confounding.

Comment about editing: Consider complying with PLOS ONE authors guidelines by not numbering the manuscript's sections and sub-sections.

Edits proposed in lines 123-125, 141-142, 178, 188, 251, 262,Table 1.

Further details about my comments and edits are available in the document attached to this decision letter.

Mickael Essouma

---

## [Author Response · Author response to Decision Letter 4]

24 Jun 2025

Dear editor Mickael Essouma

We sincerely thank you for the valuable and constructive suggestions, especially the meticulous word-for-word revisions. We have carefully studied all comments and incorporated them thoroughly into our revisions. We are very grateful for the editor's dedicated support and hope the revised manuscript is now satisfactory. Should you have any questions, please feel free to contact us.

Response: We have checked the references one by one, and no retracted article was listed in the manuscript. [22-24] and [50] were added in the references.

Additional Editor Comments:

2. The authors have continued to improve upon their manuscript, but need to polish the presentation of the materials and methods section and further edit the manuscript to improve the language quality.

Response: Thank you for your valuable feedback. We appreciate your suggestion to enhance the presentation of the materials and methods section. We will carefully review and revise this section to improve clarity, organization, and detail, ensuring that the methodology is clearly described and easily reproducible. Additionally, we will conduct a thorough language review throughout the manuscript to improve the overall quality and readability. We are committed to refining our manuscript accordingly and believe these revisions will strengthen the clarity and impact of our study.

Materials and Methods section.

3. Lines 159-164: consider addressing my request to provide references supporting the classification made for sleep duration, hypertension and diabetes.

Response: We thank the editor for your constructive suggestions. We have supplemented the following references to support the classification regarding sleep duration, hypertension and diabetes.

22. Chaput JP, McNeil J, Despres JP, Bouchard C, Tremblay A. Seven to eight hours of sleep a night is associated with a lower prevalence of the metabolic syndrome and reduced overall cardiometabolic risk in adults. PLoS One. 2013;8(9):e72832. Epub 2013/09/17. doi: 10.1371/journal.pone.0072832. PubMed PMID: 24039808; PubMed Central PMCID: PMCPMC3764138.

23. Muntner P, Hardy ST, Fine LJ, Jaeger BC, Wozniak G, Levitan EB, et al. Trends in Blood Pressure Control Among US Adults With Hypertension, 1999-2000 to 2017-2018. JAMA. 2020;324(12):1190-200. Epub 2020/09/10. doi: 10.1001/jama.2020.14545. PubMed PMID: 32902588; PubMed Central PMCID: PMCPMC7489367

24. American Diabetes A. Diagnosis and classification of diabetes mellitus. Diabetes Care. 2014;37 Suppl 1:S81-90. Epub 2013/12/21. doi: 10.2337/dc14-S081. PubMed PMID: 24357215.

4. As commented in the last round of review, you need to clearly specify that you conducted effect modification analyses for the association between 25(OH)D serum levels and PhenoAgeAccel in the statistical analysis sub-section before making the comment currently available in line 179. These references may be helpful in this respect: Edwarsd JR. Methods for Integrating Moderation and Mediation: A General Analytical Framework Using Moderated Path Analysis. Psychological Methods 2007, Vol. 12, No. 1, 1–22 and Knol et al. Recommendations for presenting analyses of effect modification and interaction. International Journal of Epidemiology 2012;41:514–520 doi:10.1093/ije/dyr218IF: 6.4 Q1 .This statement appears in the abstract: "Generalized additive models further found 25(OH)D serum 25 levels were linked to negative PhenoAgeAccel at levels below 38.2 nmol/L (15.3 ng/mL) in males (β=-0.013; 95% CI: -0.025 to -0.002) and 62.5 nmol/L (25.0 ng/mL) in females (β=-0.007; 95% CI: -0.01 to -0.004,), revealing a U-shaped relationship observed in males but an L-shaped pattern in females. . However, 25(OH)D serum levels above 125 nmol/L were not associated with PhenoAgeAccel in females (β=-0.001; 95% CI: -0.005-0.002), while in males, elevated levels (>91.55 nmol/L) were associated with positive PhenoAgeAccel (β=0.005; 95% CI: 0.002-0.008).". As requested in the last round of review, consider specifying in the methods section of the main manuscript the reasons why you chose those cut-off values for serum 25-vitamin D while performing the corresponding statistical analyses.

Response: We thank the editor for your constructive suggestions. We have supplemented moderated regression analyses incorporating a multiplicative interaction term (25(OH)D × gender) to evaluate whether the relationship between vitamin D status and biological aging differs by gender. The results of these analyses revealed statistically significant gender-specific effects (p for interaction <0.001), indicating that the association between 25(OH)D and PhenoAgeAccel is moderated by gender. In the revised manuscript, we have supplemented the following sentences in the Methods section: “To evaluate potential effect modification by gender and other key covariates, subgroup analyses and interaction tests were conducted. Interaction term of serum 25(OH)D and stratifying factor were included in the multivariable linear regression models to test for significance of interaction.”

In the Results section, we have supplemented the following sentences: “Interaction test confirmed that association between serum 25(OH)D and PhenoAgeAccel was modified by gender, regardless of whether 25(OH)D was modeled as a continuous variable or categorized using quartiles or IOM guidelines (all p for interaction <0.001).”

Regarding the cut-off values identified in the segmented regression analysis, these were determined based on data-driven statistical criteria. Specifically, we first used a generalized additive model (GAM) with a smooth term for serum 25(OH)D levels to flexibly characterize the non-linear relationship with PhenoAgeAccel, as GAMs effectively capture complex dose-response patterns without assuming parametric forms. The GAM revealed distinct non-linear associations by sex, prompting further investigation using segmented regression—a method designed to formally identify inflection points (thresholds) where the slope of the association changes directionally. For each gender, optimal breakpoints were selected via likelihood ratio tests comparing nested models, ensuring statistical rigor in threshold estimation. These thresholds were computationally derived rather than manually assigned. However, we recognized potential ambiguity in how these thresholds were presented. To clarify, we have revised the following sentences in the Abstract: “25(OH)D serum levels were negatively associated with PhenoAgeAccel (β=-0.04 standard deviation [SD]; 95% confidence interval [CI]: -0.08 to 0.00). This negative association was dose-dependent in females (β=-0.07 SD; 95% CI: -0.12 to 0.01), but not in males. Generalized additive models further revealed gender-specific non-linear patterns: a U-shape pattern in males but an L-shaped pattern in females. Using segmented regression to confirm inflection points, we observed that 25(OH)D serum levels were linked to reduced PhenoAgeAccel at levels below 38.2 nmol/L (15.3 ng/mL) in males (β=-0.013 SD; 95% CI: -0.025 to -0.002) and 62.5 nmol/L (25.0 ng/mL) in females (β=-0.007 SD; 95% CI: -0.01 to -0.004,). However, 25(OH)D serum levels above 125 nmol/L showed no association with PhenoAgeAccel in females (β=-0.001 SD; 95% CI: -0.005-0.002), while in males, elevated levels (>91.55 nmol/L) were associated with increased PhenoAgeAccel (β=0.005 SD; 95% CI: 0.002-0.008).” Additionally, we have polished the language in the Results section to improve clarity.

5. Results section. As commented in the last round of review, consider rounding all the numbers to one decimal place when providing results throughout the manuscript (Table 1 and so on).

Response: We have corrected all the numbers to one decimal place.

Comment appended to Table 1.

6. Discussion section: limitations statement. You also need to address the lack of data on vitamin D supplement among the confounders, the effect of unmeasured confounding and the reverse causation and residual confounding.

Response: This information has been added in section of limitation.

7. Comment about editing: Consider complying with PLOS ONE authors guidelines by not numbering the manuscript's sections and sub-sections.

Edits proposed in lines 123-125, 141-142, 178, 188, 251, 262,Table 1.

Response: We have made corrections following the editor’s suggestion for our manuscript. Thank you for your valuable feedback.

---

## [Editor Report · Decision Letter 4]

26 Jun 2025

Association of 25(OH)D serum level with biological aging: A Cross-Sectional Study of 2007-2016 NHANES surveys

PONE-D-24-25180R4

Dear Dr. Wang,

We’re pleased to inform you that your manuscript has been judged scientifically suitable for publication and will be formally accepted for publication once it meets all outstanding technical requirements.

Kind regards,

Mickael Essouma, M. D.

Academic Editor

PLOS ONE

Additional Editor Comments (optional):

Congratulations to the authors for substantially improving their manuscript. I have appended some last minute edits that were missed by us during the review process and can be implemented during the correction of the proof. See the document PONE-D-24-25180_R4_reviewed by Mickael Essouma.

Mickael Essouma
---

## [Editor Report · Acceptance letter]

PONE-D-24-25180R4

PLOS ONE

Dear Dr. Wang,

I'm pleased to inform you that your manuscript has been deemed suitable for publication in PLOS ONE. Congratulations! Your manuscript is now being handed over to our production team.

Kind regards,

on behalf of

Dr. Mickael Essouma

Academic Editor

PLOS ONE